# Methylation and PTEN activation in dental pulp mesenchymal stem cells promotes osteogenesis and reduces oncogenesis

Wen-Ching Shen[1], Yung-Chih Lai[2], Ling-Hui Li[3], Kolin Liao[1], Hung-Chang Lai[4], Shou-Yen Kao[4,5], John Wang[6], Cheng-Ming Chuong[2,7] & Shih-Chieh Hung [1,2,3]

Lineage commitment and tumorigenesis, traits distinguishing stem cells, have not been well characterized and compared in mesenchymal stem cells derived from human dental pulp (DP-MSCs) and bone marrow (BM-MSCs). Here, we report DP-MSCs exhibit increased osteogenic potential, possess decreased adipogenic potential, form dentin pulp-like complexes, and are resistant to oncogenic transformation when compared to BM-MSCs. Genome-wide RNA-seq and differential expression analysis reveal differences in adipocyte and osteoblast differentiation pathways, bone marrow neoplasm pathway, and PTEN/PI3K/ AKT pathway. Higher PTEN expression in DP-MSCs than in BM-MSCs is responsible for the lineage commitment and tumorigenesis differences in both cells. Additionally, the PTEN promoter in BM-MSCs exhibits higher DNA methylation levels and repressive mark H3K9Me2 enrichment when compared to DP-MSCs, which is mediated by increased DNMT3B and G9a expression, respectively. The study demonstrates how several epigenetic factors broadly affect lineage commitment and tumorigenesis, which should be considered when developing therapeutic uses of stem cells.

[1] Drug Development Center, Institute of New Drug Development, Institute of Biomedical Sciences, China Medical University, Taichung 404, Taiwan. [2] Integrative Stem Cell Center, Department of Orthopaedics, China Medical University Hospital, Taichung 404, Taiwan. [3] Institute of Biomedical Sciences, Academia Sinica, Taipei 105, Taiwan. [4] Department of Stomatology, Taipei Veterans General Hospital, Taipei 112, Taiwan. [5] Department of Dentistry, School of Dentistry, National Yang-Ming University, Taipei 112, Taiwan. [6] Department of Pathology, China Medical University Hospital, Taichung 404, Taiwan. [7] Department of Pathology, Keck School of Medicine, University of Southern California, Los Angeles, CA 90033, USA. Correspondence and requests for materials should be addressed to S.-C.H. (email: hung3340@gmail.com)

Adult stem cells, which contribute to normal regenerative processes, are believed to be cellular sources of cancer. Cell commitment/specification and tumorigenesis (i.e., uncontrolled proliferation), two main aspects of adult stem cells, are regulated by both endogenous and exogenous factors. Endogenous epigenetic factors, such as DNA methylation and chromatin modification, control adult stem cell properties[1], identities[2], and tumorigenesis[3]. Exogenous factors, such as the physical effects of the in vivo microenvironment (niche) are important in lineage specification[4,5], whereas dynamic niche signaling determines cell proliferation and fate determination of normal and cancer stem cells[6].

Human mesenchymal stem cells (MSCs), capable of self-renewal and differentiation into various cell types[7], can be isolated from various tissues including bone marrow[8] (BM-MSCs) and dental pulp (DP-MSCs)—either from adult[9] or exfoliated deciduous teeth[10]. While MSCs from different origins often exhibit cellular and molecular similarities, the distinct characteristics of MSCs from various tissues were recently delineated[11]. RNA deep sequencing (RNA-Seq) analysis of transcriptome profiles revealed the differences between BM-MSC and placenta MSC; the former increases in genes enrolled in micro-environmental process, such as bone formation and blood vessel morphogenesis, while the latter increases in genes associated with mitosis and embryonic morphogenesis[12]. Besides the differences between different tissue origins, the transcriptome profiles also differ in MSC of different commitment states[13].

Transplanting DP-MSCs into immunocompromised mice resulted in a dentin-like structure lined with human odontoblast-like cells, whereas transplanted BM-MSCs formed lamellar bone containing osteocytes and surface-lining osteoblasts, surrounding a fibrous vascular tissue with active hematopoiesis and adipocytes[9]. Besides differences in the potentials for differentiation, BM-MSCs and DP-MSCs also differ in the abilities to form tumor. Osteosarcoma with its cell-of-origin as BM-MSC accounts for the most prevalent primary bone sarcoma. However, there is no report of dental pulp sarcoma[14]. Although MSCs from different tissues recapitulate their original phenotypes; however, the machinery that regulates cell commitment and tumorigenesis remains mostly unknown and it is not clear whether a single endogenous or exogenous factor is able to link both.

The phosphatase and tensin homolog (PTEN) is a tumor suppressor that regulates multiple cell functions, including cell proliferation and differentiation[15]. Besides TP53, Rb1, CDKN2A, and cMyc, PTEN has recently been implicated as a driver of osteosarcoma[16]. Phosphatidylinositol 3-kinase (PI3K)/AKT signaling is one of the best-characterized pathways targeted by PTEN through its phosphatase activity[17]. Insulin signaling, induced by the binding of insulin or insulin-like growth factor 1 (IGF1) to the insulin receptor (IGF1R), followed by phosphorylation of the insulin receptor substrate-1 (IRS-1), which then activates the PI3K/AKT pathway[18,19], is essential for upregulating PPARγ and enhancing adipogenesis[20]. However, little is known about the role of PTEN in regulating lineage commitment and tumorigenesis in DP-MSCs and BM-MSCs. Here, we reveal that in BM-MSCs, but not in DP-MSCs, DNMT3B DNA methyltransferase and G9a histone methyltransferase establish DNA methylation and H3K9Me2 on the PTEN promoter, respectively. We further evaluate whether the repressive chromatin marks and master heterochromatin regulators have specific functions with respect to loss of PTEN expression and link lineage commitment and tumorigenesis. Thus, destabilization of specific chromosomal boundaries through DNA methylation may be a general mechanism to inactivate the PTEN tumor suppressor gene and initiate different programs in BM-MSCs than in DP-MSCs.

## Results

**DP-MSC has higher osteogenic but lower adipogenic potential.** DP-MSCs isolated from human exfoliated deciduous teeth were first examined for their stem cell properties. Approximately 15–30 adherent cells were observed 1 day after seeding on plates and derived colonies became evident after around 9 days of culture (Supplementary Fig. 1A). Although both DP-MSCs and BM-MSCs adopted fibroblast-like morphologies, the DP-MSCs were smaller than BM-MSCs at the same passage (Supplementary Fig. 1B). Flow cytometric analysis revealed that DP-MSCs were negative for CD34 and CD45 but positive for CD29, CD44, CD73, CD90, CD105, and CD166 (Supplementary Fig. 1C) suggesting that DP-MSCs displayed similar immunophenotypes as BM-MSCs, as previously assayed[21]. DP-MSCs differentiated into Alizarin Red S (ARS)-positive nodules 2 weeks after osteogenic differentiation, and expressed osteogenic markers, RUNX2 and osteocalcin (OC), 7 days after osteogenic differentiation (Supplementary Fig. 1D). As for adipogenic differentiation of DP-MSCs, Oil Red O-positive lipid clusters as well as the expression of adipogenic markers, PPARγ2 and LPL, were detected 5 weeks after inducing differentiation (Supplementary Fig. 1E). After 3 weeks of pellet culture in a chondrogenic induction medium, DP-MSCs were positive for Alcian blue staining and type II collagen (COL2A1) immune-staining, and expressed chondrogenic markers, AGGRECAN and COL2A1 (Supplementary Fig. 1F). Together, these data indicated that isolated DP-MSCs possess the characteristics of MSCs.

DP-MSCs show distinct osteogenic and adipogenic potentials from those of BM-MSCs. To clarify this, we compared the adipogenic and osteogenic potential of DP-MSCs and BM-MSCs. Nile red staining followed by the quantification revealed that fat droplet accumulation was detected in BM-MSCs 1 week after differentiation, but could not be observed in DP-MSCs until 5 weeks after differentiation (Fig. 1a and Supplementary Fig. 1E). Quantitative reverse transcription (RT)-polymerase chain reaction (PCR) further showed that BM-MSCs expressed more adipogenic markers than DP-MSCs 7 days after differentiation (Fig. 1b). In contrast, Arsenazo III staining and quantification of calcium content relative to DNA contents revealed that BM-MSCs had less calcium deposition than DP-MSCs 14 and 21 days after osteogenic differentiation (Fig. 1c). Moreover, quantitative RT-PCR showed that BM-MSCs expressed fewer osteogenic markers than DP-MSCs 1, 3, and 7 days after differentiation (Fig. 1d). Together, these data suggest that DP-MSCs have preferential osteogenic potential, while BM-MSCs undergo adipogenic differentiation in a more efficient manner.

**DP-MSC and BM-MSC show differences in PTEN signaling.** Transcriptome analysis by RNA-seq followed by principal component analyses (PCA) showed that each kind of cell clustered together (Fig. 2a). There were 3064 differentially expressed genes (p-value with FDR correction < 0.05), including 1359 upregulated and 1705 downregulated genes in BM-MSCs, compared to DP-MSCs (Supplementary Data 1). Hierarchical cluster analysis revealed a different hierarchical clustering algorithm in these cells (Fig. 2b). Among them, 102 genes were related to adipocyte differentiation (p-value = $8.65 \times 10^{-17}$), 152 genes were associated with bone cell differentiation (p-value = $5.08 \times 10^{-20}$), and 405 genes were defined as bone marrow neoplasm (p-value = $6.05 \times 10^{-18}$) by Ingenuity Pathways Analysis (IPA) (Supplementary Table 1). Differentially expressed genes between BM-MSCs and DP-MSCs imply that their capacities of cell differentiation and carcinogenesis are diverse. Furthermore, GSEA demonstrated that the PTEN pathway was significantly downregulated in BM-MSCs compared to DP-MSCs (Fig. 2c, d). PTEN

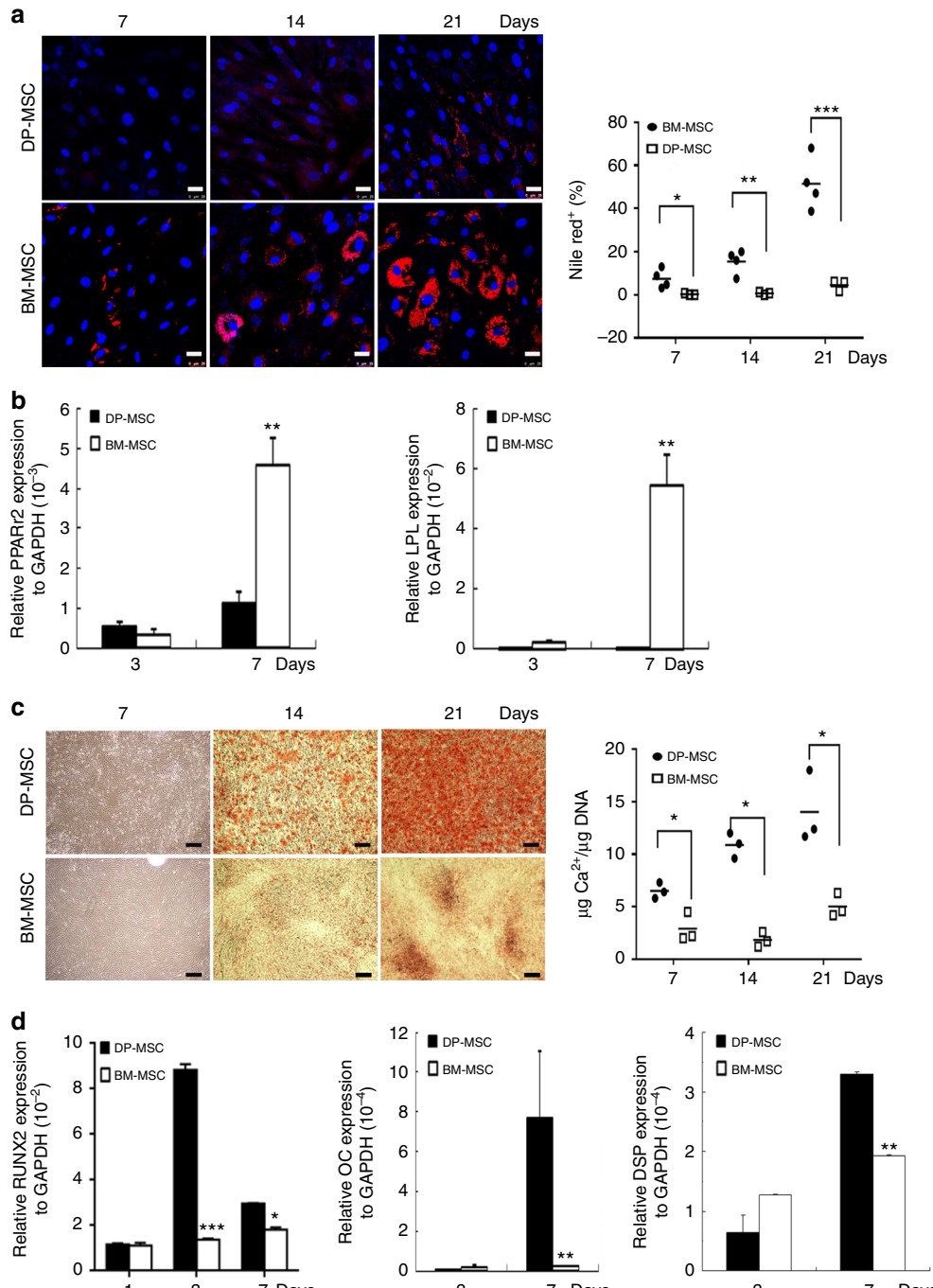

**Fig. 1** DP-MSCs decrease in adipogenesis and increases in osteogenesis. Comparison of adipogenic (**a**, **b**) and osteogenic potential (**c**, **d**) between DP-MSCs and BM-MSCs. **a** Cells were induced in an adipogenic induction medium for up to 21 days. (Left) Cells were stained by nile red, and (Right) quantification of the nile red positive cell percentages. Bar = 25 μm. **b** Quantitative RT-PCR for mRNA levels at indicated time of adipogenic differentiation. **c** Cells were induced in an osteogenic induction medium for up to 21 days. (Left) Cells were analyzed by Arsenazo III kit, and (Right) the calcium contents were quantified relative to DNA contents. **d** Quantitative RT-PCR for mRNA levels at indicated time of osteogenic differentiation. Results are expressed as the mean ± SD of three independent experiments. $*p < 0.05$; $**p < 0.01$ versus DP determined by Student's $t$-test. Bar = 200 μm

itself was downregulated in BM-MSCs compared to DP-MSCs ($p$-value with FDR correction = 0.0008, fold change = −2.50). There were 582 genes in the union of differentially expressed genes associated with adipocyte differentiation, bone cell differentiation, and bone marrow neoplasm (Supplementary Table 1), among which 272 genes were downstream targets of PTEN pathway as analyzed by using IPA Path Explorer (Fig. 2e). Several molecules in PTEN signaling seem to be regulatory hubs, e.g., PTEN and EGFR. Therefore, our results implied that the PTEN pathway plays a critical role in regulating the differential abilities of cell differentiation and carcinogenesis between BM-MSCs and DP-MSCs.

**Differences in insulin signaling and the PI3K/AKT pathway.** To confirm PTEN pathway upregulation in DP-MSCs, the PTEN level, the PTEN-targeted insulin signaling, and its downstream molecule AKT were explored. Western blotting revealed that

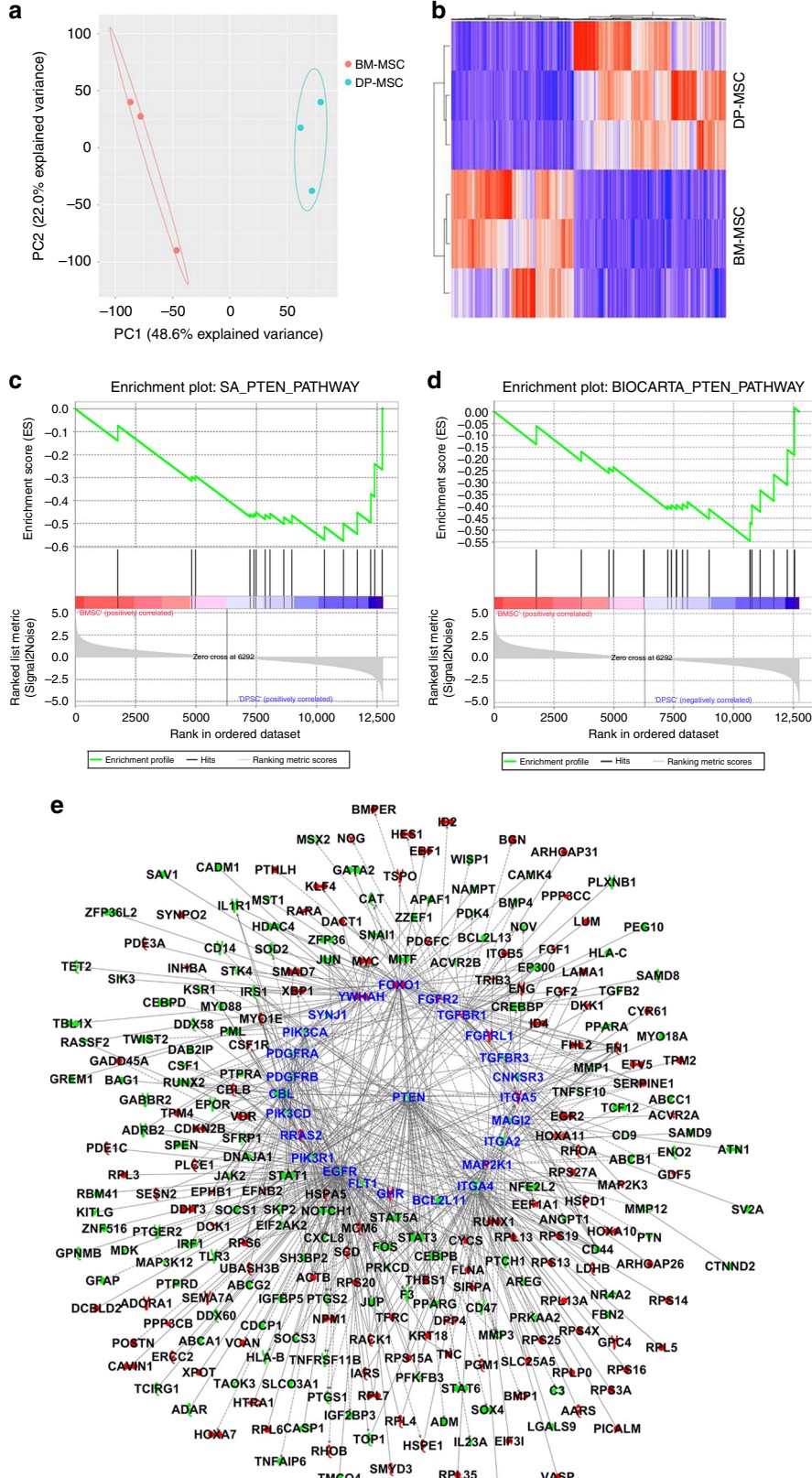

**Fig. 2** PTEN pathway was significantly upregulated in DP-MSCs. **a** PCA analysis of the global genes. **b** Heat map and hierarchical clustering analysis of differentially expressed genes. Both **c** SA_PTEN_PATHWAY (enrichment score = −0.58, p-value = 0.013) and **d** BIOCARTA_PTEN_PATHWAY (enrichment score = −0.55, p-value = 0.037) are down-regulated in BM-MSCs compared to DP-MSCs as validated by GSEA. **e** PTEN pathway plays a key upstream regulator orchestrating differentiation of bone cells and adipocytes, and bone marrow carcinogenesis. All genes in this network are differentially expressed between BM-MSCs and DP-MSCs (p-value with FDR correction < 0.05). Red nodes are genes up-regulated in BM-MSCs compared to DP-MSCs, and green nodes are genes down-regulated in BM-MSCs compared to DP-MSCs. 25 PTEN signaling genes are highlighted with red edges and increased word size

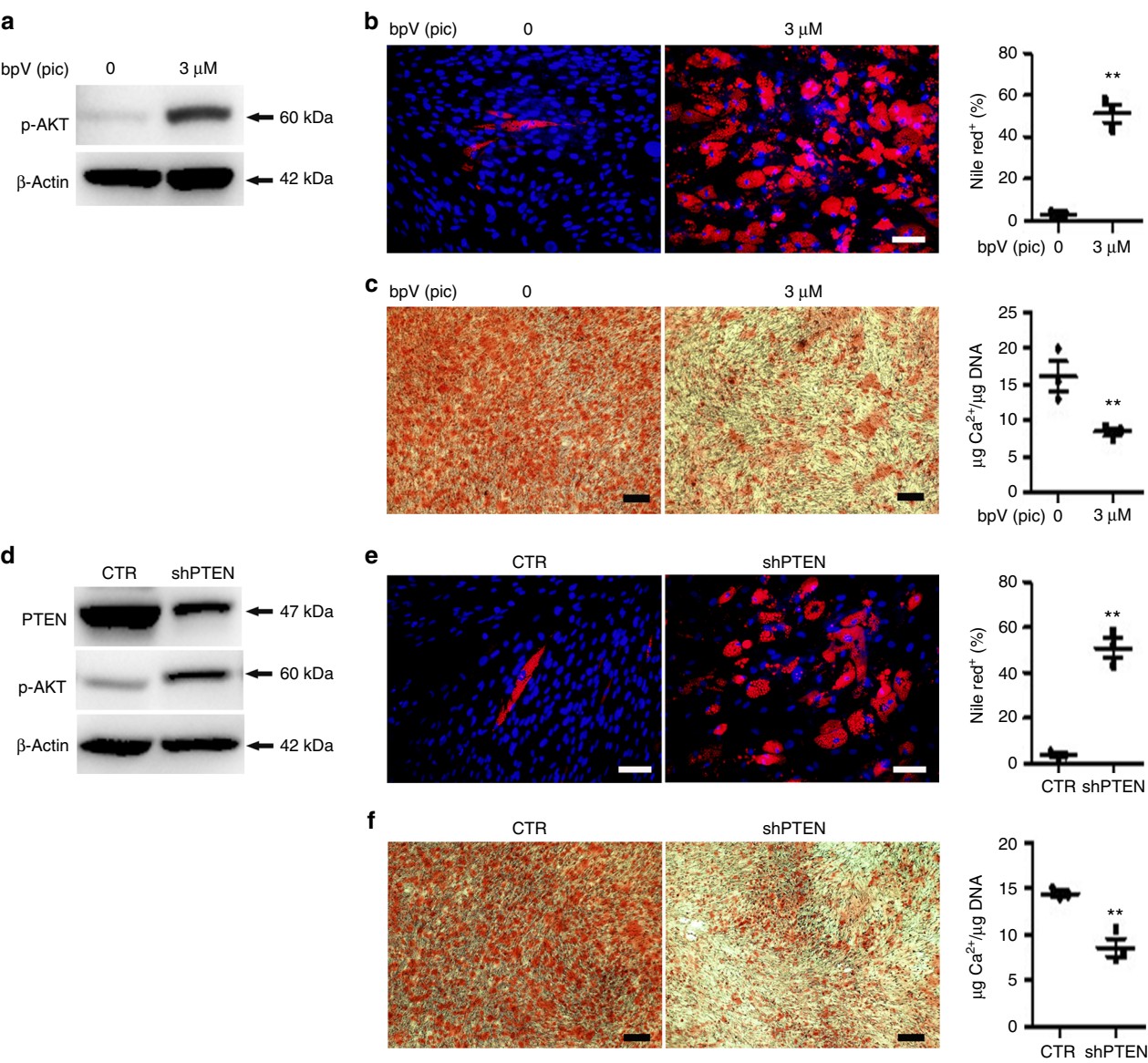

**Fig. 3** PTEN inhibition or knockdown increases adipogenesis but decreases osteogenesis in DP-MSCs. **a–c** DP-MSCs were pretreated without or with PTEN inhibitor, bpV(pic), for 2 h, followed by induction in **b** adipogenic or **c** osteogenic induction medium. **a** Cells in serum-free culture were analyzed by western blotting. Cells were analyzed by **b** nile red at 3 weeks, and **c** Arsenazo III at 2 weeks. **d–f** DP-MSCs without (CTR) or with PTEN knockdown (shPTEN) were induced in **e** adipogenic or **f** osteogenic induction medium. **d** Cells in serum-free culture were analyzed by western blotting. Cells were analyzed by **e** nile red at 3 weeks and **f** Arsenazo III at 2 weeks. **b**, **e** (right) Nile red positive cell percentages were quantified. **c**, **f** (right) The calcium contents were quantified relative to DNA contents. Quantification data are expressed as the mean ± SD of three independent experiments. *$p < 0.05$; **$p < 0.01$ versus bpV(pic) = 0 or CTR determined by Student's $t$-test. Bar = 50 μm in nile red staining. Bar = 200 μm in Arsenazo III staining

increased active AKT was initially observed within the first hour and peaked at the second hour of adipogenic induction in both DP-MSCs and BM-MSCs while the levels of active AKT were higher in BM-MSCs than in DP-MSCs (Supplementary Fig. 2A). PTEN levels were consistently higher in DP-MSCs compared to that in BM-MSCs (Supplementary Fig. 2A). Interestingly, IGF1R-IRS-1 activation was also higher in DP-MSCs compared to BM-MSCs (Supplementary Fig. 2A), suggesting a negative feedback mechanism between AKT and IGF1R-IRS-1 activation. Similar results were also observed in MSCs and DP-MSCs isolated from different individuals (Supplementary Fig. 3). Treatment of DP-MSCs and BM-MSCs induced for adipogenesis with LY294002, a specific PI3K inhibitor, inactivated AKT (Supplementary Fig. 2B), suppressed the expression of adipogenic genes (Supplementary Fig. 2C), and decreased Oil Red O staining (Supplementary

Fig. 2D). Moreover, this suppression was more pronounced in BM-MSCs than in DP-MSCs. These data suggested a role of PI3K/AKT signaling in lineage commitment during adipogenesis.

**PTEN defines differential lineage commitment**. We then examined the role of PTEN in regulating lineage commitment. In DP-MSCs, PTEN inhibition by bpV(pic) treatment or PTEN knockdown activated AKT, (Fig. 3a, d), enhanced nile red staining and increased nile red positive cell percentages 21 days after differentiation (Fig. 3b, e), and increased the expression of adipogenic genes 7 days after adipogenic differentiation (Supplementary Fig. 4A and C). In contrast, PTEN inhibition or knockdown suppressed Arsenazo III staining and calcium contents relative to DNA contents 14 days after osteogenic

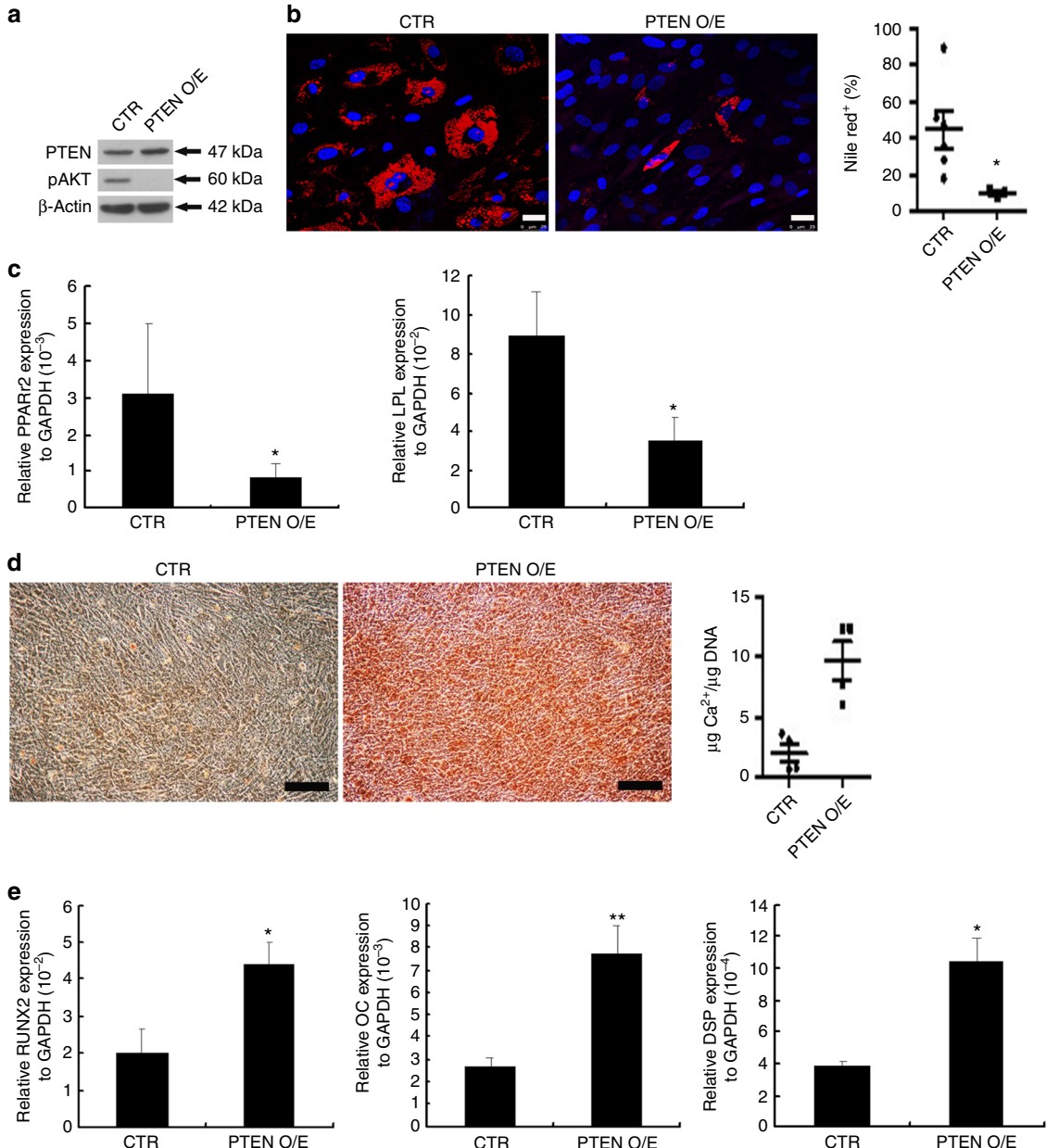

**Fig. 4** PTEN overexpression decreases adipogenesis and increases osteogenesis in BM-MSCs. **a–e** BM-MSCs were transfected with a control plasmid pcDNA6™V5-HisA (CTR) or an overexpression plasmid, pcDNA6-PTEN (PTEN O/E), followed by induction for **b**, **c** adipogenic or **d**, **e** osteogenic differentiation. **a** Cells were analyzed by western blotting. **b** Cells induced for adipogenic induction were stained by nile red and (right) quantification of nile red positive cell percentages. Bar = 25 μm. **c** Quantitative RT-PCR for the mRNA levels 7 days after differentiation. **d** Cells induced for osteogenic differentiation were analyzed by Arsenazo III kit and (right) quantification of the calcium contents relative to DNA contents at 2 weeks. Bar = 200 μm. **e** Quantitative RT-PCR for the mRNA levels 7 days after differentiation. **c**, **e** Results are shown as the relative expression to GAPDH (mean ± SD of three independent experiments), and significance was determined by Student's *t*-test. (*$p < 0.05$ and **$p < 0.01$ versus CTR)

differentiation (Fig. 3c, f), and inhibited the expression of osteogenic and odontogenic genes 7 days after osteogenic differentiation (Supplementary Fig. 4B and D).

In BM-MSCs, PTEN overexpression inactivated AKT (Fig. 4a), decreased nile red positive cell percentage 14 days after adipogenic differentiation (Fig. 4b), and suppressed adipogenic gene expression 7 days after adipogenic induction (Fig. 4c). Moreover, PTEN overexpression enhanced Arsenazo III staining and calcium contents relative to DNA contents 2 weeks after osteogenic induction (Fig. 4d), and upregulated the expression of osteogenic and odontogenic genes 7 days after osteogenic differentiation (Fig. 4e). These results suggested that PTEN is a

key regulator in controlling lineage commitment during adipogenesis and osteogenesis.

**PTEN methylation mediated by DNMT3B in BM-MSC.** DNA methylation plays an important role in controlling PTEN expression in a variety of tumor cells[22]. We therefore investigated the involvement of PTEN DNA methylation in lineage commitment in DP-MSCs and BM-MSCs. Methylation-specific PCR revealed that DNA methylation in the PTEN promoter region was upregulated in all tested BM-MSCs compared with DP-MSCs (Fig. 5a). Treatment with a DNA methylation inhibitor, 5-aza-2′-

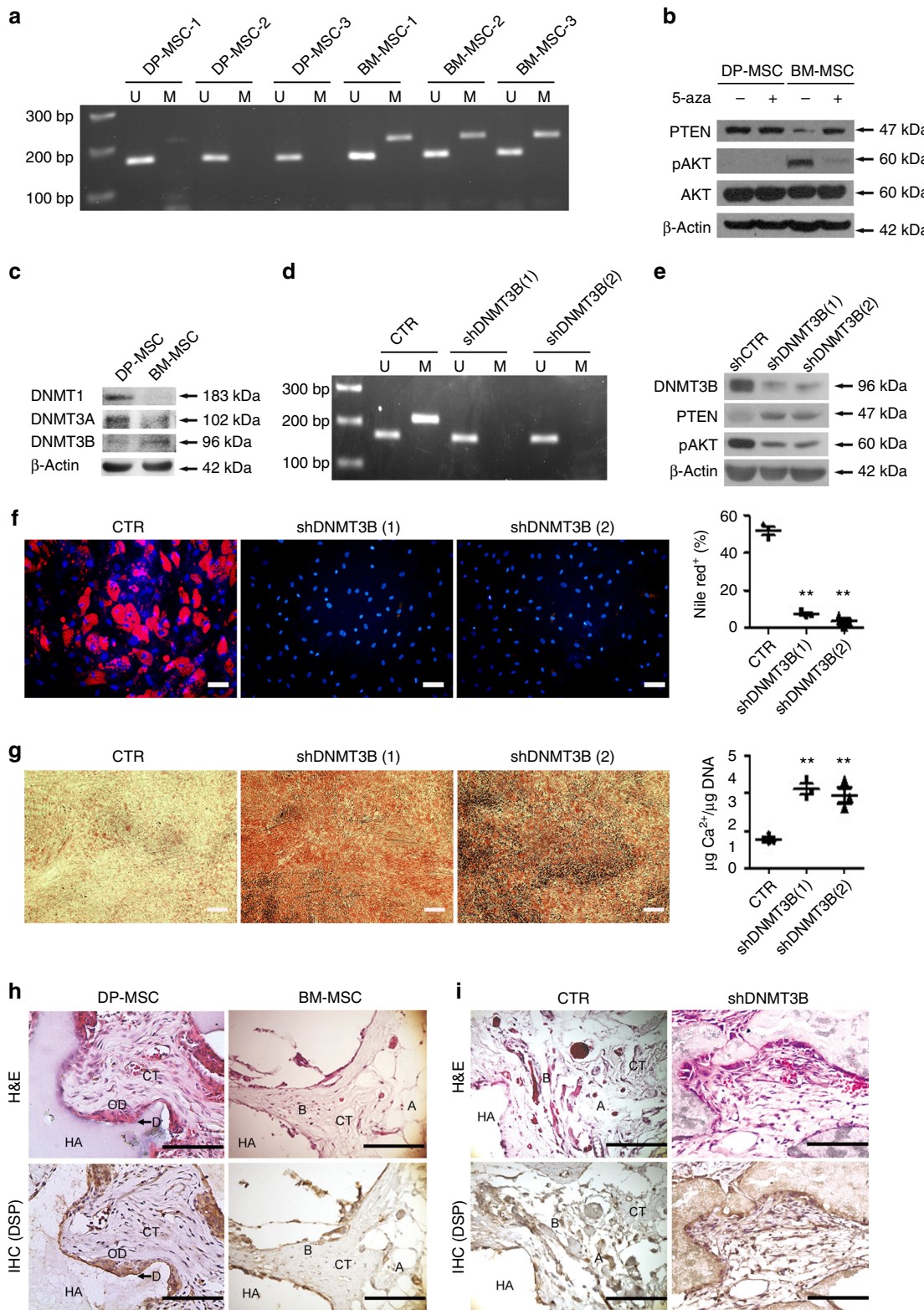

deoxycytidine (5-aza) for 4 days significantly increased PTEN expression and decreased AKT phosphorylation in BM-MSCs, but not in DP-MSCs (Fig. 5b) suggesting that PTEN DNA methylation is a key factor mediating the differential PTEN expression between DP-MSCs and BM-MSCs. Western blotting analysis of the expression of DNA (cytosine-5) methyltransferase (DNMT) family members showed that the expression of DNMT3B, rather than DNMT1 and DNMT3A, was higher in

BM-MSCs than in DP-MSCs (Fig. 5c). Knocking down DNMT3B in BM-MSCs decreased the level of DNA methylation at the PTEN promoter (Fig. 5d), upregulated PTEN expression, and decreased AKT phosphorylation (Fig. 5e). Furthermore, as expected, DNMT3B knockdown inhibited adipogenic potential (Fig. 5f) and increased osteogenic potential in BM-MSCs (Fig. 5g). DP-MSCs form a dentin pulp-like complex after subcutaneous transplantation in vivo[9]. In agreement with this

**Fig. 5** DNMT3B suppresses PTEN expression to promote adipogenesis and inhibit osteogenesis. DP-MSCs and BM-MSCs were separately isolated from three different individuals. **a** Methylation-specific PCR of PTEN. U, fragments amplified by unmethylated sequence-specific primers; M, fragments amplified by methylated sequence-specific primers. **b** Western blotting analysis of cells treated without or with 1 μM 5 aza-cytidine (5aza) for 4 days. **c** Western blotting analysis of DP-MSCs and BM-MSCs. **d**–**h** BM-MSCs were transduced with control shRNAs (CTR) or with different DNMT3B shRNAs (shDNMT3B (1) and (2)). Cells were analyzed by **d** methylation-specific PCR of PTEN and **e** western blotting. **f** Cells were induced for adipogenic induction for 3 weeks and stained by nile red staining. Right Quantification of nile red positive cell percentages at 3 weeks. Bar = 50 μm. **g** Cells were induced for osteogenic differentiation for 2 weeks and stained by Arsenazo III kit. Right Quantification of the calcium contents relative to DNA contents. Bar = 200 μm. Results are expressed as the mean ± SD of three independent experiments. **h** DP-MSCs, BM-MSCs, and **i** BM-MSCs transduced with control shRNAs (CTR) or with DNMT3B shRNAs (shDNMT3B) were delivered in HA/TCP powder (10^6 cells in 10 mg) and transplanted underneath the dorsal skin of NOD/SCID mice for 8 weeks. Sections of transplants were analyzed by H&E and immunohistochemistry of DSP. Newly formed dentin (arrows) and odontoblasts (O.D.) were immunoreactive for DSP antibody in single-colony-derived DP-MSC transplants. HA HA/TCP, D dentin, CT connective tissue, B bone, A adipose tissue. Bar = 100 μm

previous report, transplanted DP-MSCs were capable of generating abundant ectopic pulp-like complexes in an HA/TCP carrier whereas the transplanted BM-MSCs formed new bone containing osteocytes and surface-lining osteoblasts, surrounding a fibrous vascular tissue with active hematopoiesis and adipocytes (Fig. 5h). Interestingly, DNMT3B knockdown induced transplanted BM-MSCs to produce dentin pulp-like complexes (Fig. 5i). Immunohistochemical study further demonstrated that DP-MSCs and BM-MSCs with DNMT3B knockdown, but not normal BM-MSCs, expressed human DSP. These data indicated that PTEN DNA methylation mediated by DNMT3B is a key regulator in DP-MSC and BM-MSC lineage commitment.

**Lysine methyltransferase G9a suppresses PTEN to activate AKT.** A pronounced increase in histone H3 methylation on Lys 9[23] or Lys 27[24] is a prerequisite step for de novo methylation at the promoter by the enzymes DNMT3A/3B in early embryonic or cancer genes. A chromatin immunoprecipitation (ChIP) assay revealed that BM-MSCs exhibited increased enrichment of DNMT3B and histone H3-lysine 9 dimethylation (H3K9me2) but not histone H3-lysine 27 trimethylation (H3K27me3) in the PTEN promoter region (at sites 123–329 bp upstream of transcriptional start site) when compared to DP-MSCs (Fig. 6a). Lysine methyltransferase G9a operates as a master regulator that inactivates numerous early-embryonic genes by bringing about heterochromatinization of methylated histone H3K9[24,25]. Western blotting further revealed that BM-MSCs showed an increase in the protein levels of H3K9me2 and G9a rather than H3K27me3 or EZH2 (Fig. 6b, c). Immunofluorescence also revealed BM-MSCs increased co-localization of H3K9me2 and G9a in the nucleus compared to DP-MSCs (Fig. 6d). Moreover, knockdown of G9a in BM-MSCs upregulated PTEN expression, decreased AKT phosphorylation (Fig. 6e), inhibited adipogenesis (Fig. 6f), and promoted osteogenesis (Fig. 6g). Together, these data suggested that lysine methyltransferase G9a is required for DNMT3B-mediated PTEN suppression, which activates AKT to promote adipogenesis and inhibit osteogenesis.

**Linking lineage commitment with tumorigenesis.** MSCs reside in bones and teeth. However, osteosarcoma, the most common primary bone malignancy, always occurs in bone but never in teeth. Moreover, the worldwide incidence of malignant odontogenic tumors is very rare[26]. These studies suggest that BM-MSCs and DP-MSCs exhibit differences in tumorigenic potential. A combination of retinoblastoma (Rb) silencing and cMyc overexpression has successfully transformed human MSCs into malignant cells[27]. We demonstrated that BM-MSCs from three individuals were successfully transformed with this protocol (Fig. 7a), while DP-MSCs were only transformed when combined with PTEN silencing. Interestingly, transformed BM-MSCs

(SiRb-OeMyc) formed osteosarcoma-like tumors in immunodeficient mice (Fig. 7b), while transformed DP-MSCs (SiRb-OeMyc-SiPTEN) formed odontogenic tumors (Fig. 7c, d) associated with osteoid (Fig. 7e) and adipose tissues (Fig. 7c). To support our findings, we analyzed human RNA-Seq data of 3 normal bone samples, 44 osteosarcomas patient samples, and 5 osteosarcomas cell lines from Gene Expression Omnibus (GEO) accession GSE87624[28]. We found that PTEN levels are significantly different among the three groups (Supplementary Fig. 5, ANOVA, $F_{2,49} = 9.576$, p-value = 0.0003). Using normal bone samples as the control, PTEN levels are significantly downregulated in human patient osteosarcomas samples (Tukey HSD post-hoc test, p-value = 0.0196). PTEN levels in human osteosarcoma cell lines are more significantly down-regulated compared to normal bones (Tukey HSD post-hoc test, p-value = 0.0002). These comparisons support that PTEN levels were deregulated in human osteosarcomas. These data suggest that PTEN expression differentiates tumorigenic potential between BM-MSCs and DP-MSCs.

**Discussion**
Previous studies showed that DP-MSCs and BM-MSCs shared similarities in morphology, plastic adherence, surface antigens, and gene profiles, whereas more recent investigations argued that variations in proliferation and differentiation potentials could be distinguished between these two cells[9,10,29,30], which may be due to their different ontology and differential developmental potential in vivo[10]. In the current study, DP-MSCs exhibited increased osteogenic and odontogenic potential, but less adipogenic potential, and showed resistance to oncogenic transformation by SiRb-OeMyc in comparison with BM-MSCs. Transcriptome analysis revealed that DP-MSCs and BM-MSCs differentially express genes associated with adipogenesis, osteogenesis, carcinogenesis, and the PTEN pathway. Differential PTEN expression, mediated by PTEN methylation, was responsible for the lineage commitment and tumorigenesis differences in both cells. Further molecular analysis demonstrated that BM-MSCs had a higher DNA methylation level and repressive mark H3K9Me2 enrichment at the PTEN promoter compared to those of DP-MSCs, which were mediated by increased DNMT3B and G9a levels, respectively. The results support the roles of the epigenetic factors in PTEN expression and in regulating MSC lineage commitment and tumorigenesis.

PTEN expression regulated by DNA methylation was only reported to be involved in tumor progression and chemotherapy resistance[31–33]. It was shown that PTEN promoter hypermethylation suppresses PTEN expression and activates AKT in tamoxifen-resisted breast cancer cells, and treatment with 5-aza reduces PTEN methylation and sensitizes cells to tamoxifen-induced cytotoxicity[31]. Although PTEN has been shown to be important for regulating proliferation and differentiation of

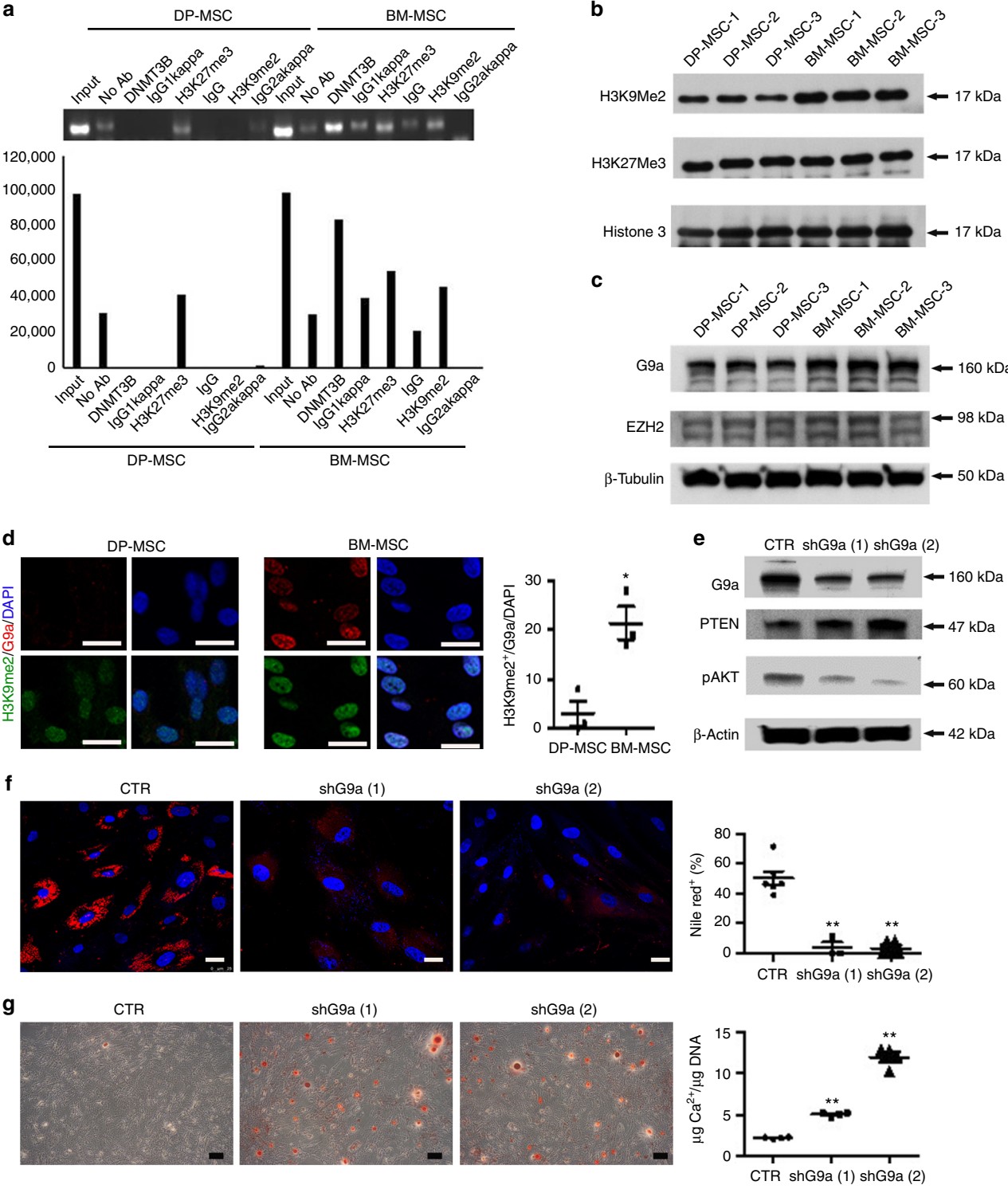

**Fig. 6** G9a suppresses PTEN expression to promote adipogenesis and inhibit osteogenesis. DNMT3B and H3K9me2 are abundant at the promoter of upstream 123–329 bp PTEN locus in BM-MSCs compared with DP-MSCs. **a** ChIP analysis for determining the occupancy of PTEN promoter by DMNT3B and repressive marks. Below: Quantification of the results. Western blotting (**b**, **c**) and immunofluorescence (**d**) showed that BM-MSCs increased in the protein level of H3K9Me2 and master regulator, G9a. Bar = 25 μm. **e** Lentiviral knockdown of G9a in BM-MSCs increased PTEN and decreased AKP phosphorylation. **f** G9a knockdown cells were induced for adipogenic differentiation for 2 weeks and stained by nile red staining. Bar = 25 μm. Right Quantification of nile red positive cell percentages. **g** The cells were also induced for osteogenic differentiation for 3 weeks and stained by Arsenazo III kit. Bar = 200 μm. Right Quantification of the calcium contents relative to DNA contents. Results are expressed as the mean ± SD of three independent experiments

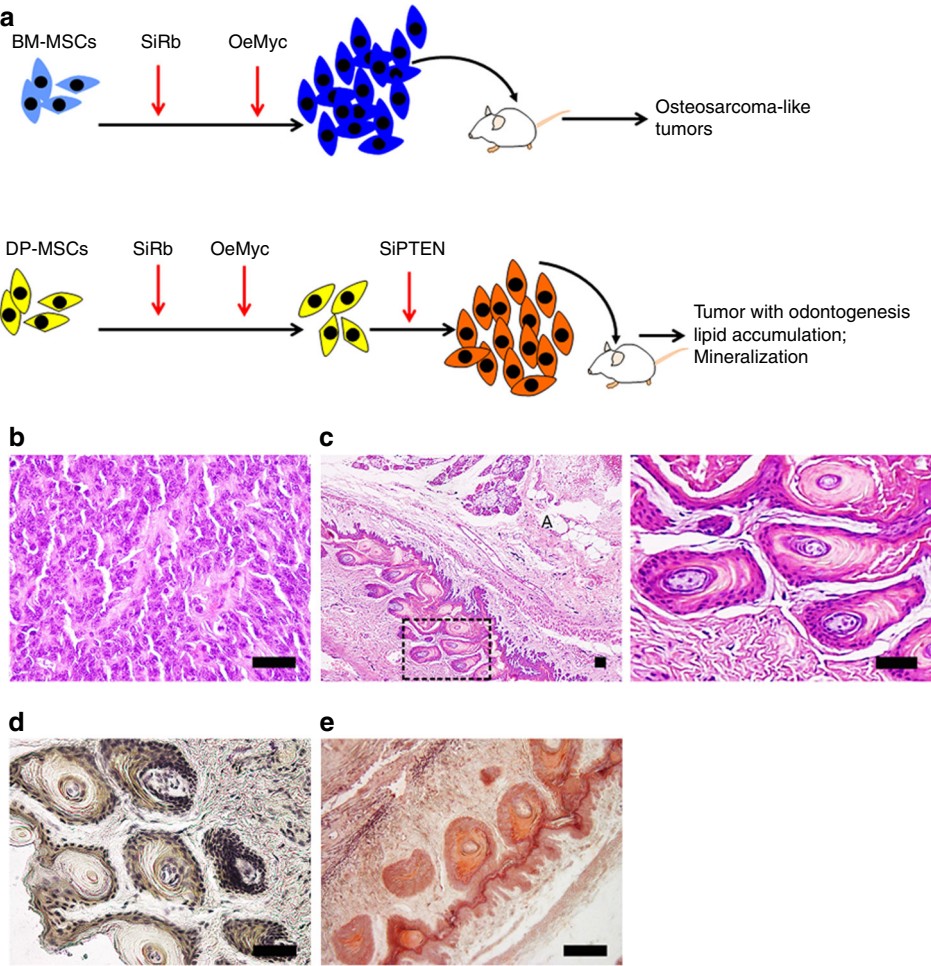

**Fig. 7** PTEN silencing increases tumorigenesis. **a** Flow chart showing the protocols for oncogenic transformation of BM-MSCs and DP-MSCs. Transformed BM-MSCs (SiRb-OeMyc) formed osteosarcoma-like tumors in immunodeficient mice (**b**), while transformed DPSCs (SiRb-OeMyc-SiPTEN) formed odontogenic tumors associated with osteoid and adipose tissues (**c–e**). Sections of transplants were analyzed by H&E (**b**, **c**), immunohistochemistry of DSP (**d**), and Alizarin Red S (**e**). A in (**c**): adipose tissue. **c**, right The rectangular area is shown at higher magnification. Bar = 50 μm. SiRb: knockdown of Rb; OeMyc: overexpression of cMyc; siPTEN: knockdown of PTEN

mouse auditory progenitors[34] and zebrafish hematopoietic stem cells[35]; to our best understanding, the present study represented the first evidence that PTEN expression regulated by DNA methylation serves as a key player to control lineage commitment and tumorigenesis in human adult stem cells. However, PTEN is not a molecular or functional signature of BM-MSCs when compared to placenta MSCs, adipose MSCs, hematopoietic stem and progenitor cells, skin fibroblasts, or osteoblasts[12,13]. We believe that the characteristics of DP-MSCs, including extremely low tumorigenic potential[14] and unique cell fate, are critical for the development of novel functional signatures between DP-MSCs and BM-MSCs.

In addition, the role of DNMT3B was demonstrated in regulating phenotypic and molecular differences between MSCs from different origins. Interestingly, PTEN methylation controlled by DNMT3A, rather than DNMT3B, was previously reported in hepatocellular carcinomas[36] indicating differential cell-type specific regulation of PTEN methylation by DNMT proteins. The in vivo transplantation assays showing that DNMT3B knockdown enhanced the differentiation potential of BM-MSCs to form a pulp-like complex further provided evidence to support the possibility of converting BM-MSCs into DP-MSCs by genetic manipulation. For the clinical application of MSCs in dental regeneration, approaches involving BM-MSC isolation, followed by conversion into DP-MSCs, may be more practicable for the elderly, who have no preserved DP-MSCs from exfoliated deciduous teeth.

G9a is essential for early embryo development and embryonic stem cell differentiation and plays a dominant role in establishing and maintaining the abundant repressive H3K9me2 modification[25,37]. The G9a-dependent H3K9me2 is also important for lineage commitment in hematopoietic stem cells[38], and suppressing adhesion molecules and promoting invasion and metastasis in lung cancer cells[39]. However, the roles of G9a and the repressive H3K9me2 modification in MSCs are not well-studied. This study implicates the role of G9a-dependent H3K9me2 in specifying lineage commitment and tumorigenesis in MSCs derived from different tissues.

As shown by a previous study using microarray analysis, DP-MSCs and BM-MSCs similarly express 4400 known genes. For those genes differentially expressed in DP-MSCs and BM-MSCs, enriched expression of collagen type XVIII α1, IGF-2, discordin domain tyrosine kinase 2 (DDR2), NADPH menadione oxidoreductase, homolog 2 of Drosophila large disk, and cyclin-dependent kinase 6 were detected in DP-MSCs but not in BMSCs[11]. While IGF-2 can activate AKT[40], elevated IGF-2 in DP-MSCs found in the previous study may explain the upregulated IGF1R and IRS-1 phosphorylation in DP-MSCs found in the current study. The activation of IGF1R and IRS-1, however, did not induce the final AKT signaling, arguing that

IGF1R-IRS-1 pathway is not a key determinant to differentiate DP-MSCs and BM-MSCs. Based on the results of the present study, increased PTEN expression in DP-MSCs can block IGF-2-mediated signals, suggesting that PTEN plays a more important role in controlling insulin signaling and AKT activation. We found that PTEN levels are deregulated in human osteosarcoma. Consistently, Xi et al. found that PTEN is reduced in human osteosarcoma cell lines as compared to normal human osteoblasts[41]. Gong et al. also found that PTEN is expressed at low level in human osteosarcomas compared to adjacent tissues[42]. Moreover, the down-regulation of PTEN enhanced migration and invasion of osteosarcoma cell lines[43] and is correlated with poor prognosis in human osteosarcomas[44,45]. Previously, Freeman et al. also summarized that the loss of PTEN was common in osteosarcoma[46]. All the above studies advocate the down-regulation of PTEN in the human osteosarcoma.

There are still limitations in the current study. Because the ages of the donors who provided the samples were not consistent between isolated DP-MSCs and BM-MSCs, the related differences may be related to the age of source of stem cells. To deal with this issue, we integrated 48 human and rhesus monkey microarray datasets downloaded from public repositories (Supplementary Data 2). Both human and rhesus monkey expression profiles manifested that PTEN expression levels in BM-MSCs are similar among different ages and developmental stages (Supplementary Fig. 6). By contrast, PTEN is significantly up-regulated in DP-MSCs compared to BM-MSCs in the same age donors, consistent with our findings (Supplementary Fig. 6A). These integrated analyses have revealed that age is not a major confounding factor in this study and further reinforced our discovery.

In conclusion, our findings demonstrated that DP-MSCs and BM-MSCs differed in their adipogenic and osteogenic potential, and reprogramming the native tissue-specific differentiation potential was achievable by modulating the G9a/DNMT3B/PTEN/AKT pathway. We believe that this molecular machinery to determine choice of specific cell fate and tumorigenesis will be of great value in regenerative medicine and future clinical uses.

## Methods

**Cell isolation and culture.** Normal exfoliated human deciduous incisors were collected from 7–8-year-old children in the Pediatric Dental Department, while bone marrow aspirates were collected from patients who received an orthopedic surgery in the orthopedic department of the Taipei Veterans General Hospital with the approval of Institute of Review Board. Informed consent was obtained from all subjects. Details regarding the donor information are shown in Supplementary Table 2. The pulp was isolated from a remnant crown and then digested in a solution of 3 mg/ml collagenase type I (Worthington Biochemical Corp., Lakeside, NJ) and 4 mg/ml dispase (Roche Molecular Biochemicals, Mannheim, Germany) at 37 °C for 1 h. The cell suspension was filtered through a 40 μm cell strainer (BD Falcon; BD Biosciences, Bedford, MA) and the nucleated cells were plated in 6-well plastic dishes at clonal density and cultured in complete medium [CM: α-MEM (α-minimal essential medium; Gibco-BRL, Gaithersburg, MD), supplemented with 10% fetal bovine serum (FBS), 100 units/ml penicillin, 100 μg/ml streptomycin, and 2 mM L-glutamine (Invitrogen, Carlsbad, CA)] at 37 °C under 5% $CO_2$ atmosphere. Human BM-MSCs have originally isolated from the iliac crest of donors[21]. Briefly, mononuclear cells were isolated from heparinized bone marrow by density gradient centrifugation using Ficoll-Hypaque of a density of 1.077 g/L, followed by seeding into 6-well plate with CM at 37 °C under 5% $CO_2$ atmosphere. At 9 days after seeding, both of DP-MSCs and BM-MSCs were recovered and reseeded in 10-cm plastic dishes at an initial density of $4 \times 10^3$ cells/cm² and the culture medium was changed twice per week. The medium was changed twice a week and a subculture is performed at 1:3 to 1:5 split every week. Mycoplasma contamination was checked every 1 month. Cells with mycoplasma contamination were discarded.

**Flow cytometric analysis.** For the flow cytometric analysis of surface proteins, adherent cell monolayer of DP-MSCs and BM-MSCs were harvested by 5 mM EDTA in phosphate-buffered saline (PBS). Cells were incubated with FITC- or PE-conjugated antibodies against human CD29, CD34, CD44, CD45, CD73, CD90, CD105, and CD166. Matched isotype antibodies were served as controls (Becton Dickinson, San Jose, CA). Cells, washed once with cold PBS containing 2% fetal calf serum. Then, thousand labeled cells were acquired and analyzed using a FACScan flow cytometry running CellQuest software (Becton Dickinson). Gating strategy for analyzing MSC flow cytometry data is shown in Supplementary Fig. 7.

**Cell differentiation protocols.** Before the initiation of differentiation, cells were seeded in growth medium at a density of 10⁴ cells/cm². For in vitro differentiation into osteoblasts, adipocytes, and chondrocytes, cells were induced with osteogenic induction medium (OIM), composed of α-MEM supplemented with 10% FBS, $10^{-8}$ M dexamethasone, 50 μg/ml ascorbic acid-2 phosphate, 10 mM β-glycerophosphate (Sigma, St. Louis, MO); adipogenic induction medium (AIM), composed of α-MEM supplemented with 10% FBS, 50 mg/ml ascorbate-2 phosphate (Sigma), $10^{-7}$ M dexamethasone (Sigma), 50 mg/ml indomethacin (Sigma), and 10 mg/ml insulin (Sigma); and chondrogenic induction medium (CIM), with cell pellets in serum-free α-MEM supplemented with, $10^{-7}$ M dexamethasone, 50 μg/ml ascorbic acid-2 phosphate (Sigma), 10 ng/ml TGF-β1 (PeproTech), ITS-Premix (GIBCO), respectively. After the appearance of morphologic features of specific lineages, cells in OIM and AIM were stained with ARS and Oil Red O, respectively. Cells in CIM were prepared for paraffin sections (4-μm in thickness) and Alcian blue staining and immunohistochemistry. For immunohistochemistry, paraffin sections were initially incubated with blocking serum, probed with a monoclonal antibody against human type II collagen (Millipore), and DAB staining (brown) (Vector Laboratories).

**Nile red staining.** For nile red staining, medium was aspirated and washed twice with PBS. Cells were then fixed in 1 ml of 10% neutral-buffered formalin for 60 min. Following fixation, cells were washed once with PBS and then incubated for 15 min with 250 μg/ml nile red (Sigma) in PBS at room temperature. The cells were counterstained with 4′,6-diamidino-2-phenylindole (DAPI), and images were captured using digital imaging (×200). Nile red positive cell percentages were counted.

**Calcium assay.** Calcium contents of cell were quantified by colorimetric endpoint assay based on the complexation of $Ca^{2+}$ with Arsenazo III molecule at 1:2 ratio to a blue-purple product. Aliquots of 100 μl of Arsenazo III solution (Pointe Scientific, Canton, MI) were added to the wells and incubated for 10 min at room temperature. After removal of the solution, aliquots of 100 μl of 0.6 M HCl were added to the wells for 2 h at room temperature. The dissolved mineral in the HCl solution was quantified spectrophotometrically at 650 nm. A standard dilution series of calcium standard (Merck, Billerica, MA) ranging from 0 to 160 μg/ml was prepared and quantified spectrophotometrically at 650 nm. The amount of DNA in the wells were determined with Picogreen fluorescent dye (Molecular Probes, Eugene, OR) according to the manufacturer's instructions. Calcium content was expressed as micrograms of $Ca^{2+}$/μg DNA.

**RT-PCR and real-time PCR.** Total RNA was extracted using the Trizol reagent (Invitrogen) according to the manufacturer's specifications. First strand cDNA synthesis was performed using Superscript III reverse transcriptase (Invitrogen), Random primer (Invitrogen), 10 mM dNTPs (Invitrogen), 5X First Strand synthesis buffer, 0.1 M DTT, and RNaseOUT ribonuclease RNase inhibitor (Invitrogen). PCR was performed using cDNA as the template in a 50 μl reaction mixture containing a specific primer pair of each cDNA according to the published sequences (Supplementary Table 3). Each cycle consisted of three steps: denaturation for 45 s at 94 °C, annealing for 1 min at 51–58 °C, and 90 s of elongation at 72 °C. The reaction products were resolved by electrophoresis on a 1.5% agarose gel and visualized with ethidium bromide. The protocol of quantitative real-time PCR was performed using cDNA as the template in a 20 μl reaction mixture containing FastStart SYBR Green Master (Roche Applied Science) and a specific primer pair of each cDNA according to the published sequences (Supplementary Table 3). Analysis of the results was carried out using the software supplied with the ABI Step One Real-Time PCR System machine and calculated expression relative to the human GAPDH and then relative to controls (delta delta CT) using the fluorescence threshold of the amplification reaction and the comparative CT method.

**Methylation-specific PCR.** The bisulfite modification of DNA samples was performed using the MethylEasy DNA Bisulfite Modification kit (Human Genetic Signatures). Methylation-specific PCR analysis for PTEN was performed following previous protocol[47]. The primer sequences are as follows: PTEN methylation (PTEN-M) primers: sense, 5′-TTCGTTCGTCGTCGTCGTATTT-3′; antisense, 5′-GCCGCTTAACTCTAAACCGCAACCG-3′; PTEN unmethylation (PTEN-U) primers: sense, 5′-GTGTTGGTGGAGGTAGTTGTTT-3′; antisense, 5′-ACCACTTAACTCTAAACCACAACCA-3′. PCR products (206 or 162 bp) were resolved on 2.5% agarose gels after ethidium bromide staining and UV transillumination. The uncropped scan of the blot is shown in the Source Data file.

**Western blot analysis.** Cells were rinsed with PBS and lysed in 0.2 ml of protein extraction reagent (M-PER, Pierce, Rockford, IL) plus protease inhibitor cocktail (HaltTM, Pierce) for 5 min on ice. Protein concentrations were determined using the BCA assay (Pierce). After being heated for 5 min at 95 °C in a sample buffer, equal aliquots of the cell lysates were run on a 10% SDS–polyacrylamide gel. Proteins were transferred to PVDF membrane filters, following by incubation in blocking buffer for 1 h. The filters were then incubated overnight at 4 °C with a

1:1000 dilution in Tris-buffered saline (0.2% Tween 20, 1.36 M NaCl, 200 mM Tris-base) of antibodies against pAKT (9271, Cell Signaling), pIRS-1 (2384, Cell Signaling), pIGF1R (3027, Cell Signaling), PTEN (9552, Cell Signaling), ß-ACTIN (4967, Cell Signaling), DNMT1 (WH0001786M1, Cell Signaling), DNMT3A (GTX104955, Gene Tex), DNMT3B (GTX62171, Gene Tex), G9a (ab40542, Abcam) and EZH2 (49-1043, Invitrogen). The filters were washed and bound primary antibodies were detected by incubating for 1 h with horseradish peroxidase-conjugated goat anti-mouse or anti-rabbit or donkey anti-goat IgG (PharMingen). The filters were washed and developed using a chemiluminescence assay. The uncropped scans of the most important blots are shown in the Source Data file.

**Plasmid construction and transfection.** Overexpression plasmid, pcDNA6-PTEN (PTEN O/E), was generated by inserting a 2357-bp fragment of the full-length human PTEN cDNA from pOBT7-PTEN (Open Biosystems, Huntsville, AL) into the EcoRI/XhoI sites of the pcDNA6$^{TM}$V5-HisA vector (Invitrogen). Transfection was carried out by Nucleofector technology (AMAXA Biosystems, Cologne, Germany) with each nucleofection sample containing 2–4 μg of DNA, $4 \times 10^5$ cells, and 100 μl of Human MSC Nucleofector Solution using the program C-17 of the Nucleofector device, as recommended by the manufacturer.

**Lentiviral-mediated RNAi.** The expression plasmids and the bacteria clone for PTEN shRNA (TRCN0000029738), DNMT3B shRNA (TRCN0000035687 and TRCN0000035688), and G9a (TRCN00000115667 and TRCN00000115668) were provided by the National Science Council in Taiwan. Lentiviral production was performed by transfection of 293T cells using Lipofectamine 2000 (LF2000; Invitrogen, Carlsbad, CA). Forty-eight hours after transfection, supernatants were collected and filtered. Subconfluent cells were infected with lentivirus in the presence of 8 μg/ml polybrene (Sigma-Aldrich). At 24 h post-infection, the medium was replaced with fresh growth medium containing puromycin (1 μg/ml) and selected for infected cells for 48 h.

**RNA-Seq.** Total RNA was isolated with the RNeasy Micro Kit (QIAGEN). Illumina TruSeq RNA sample Prep kit was used to prepare RNA-Seq library. Both BM-MSCs and DP-MSCs RNA-Seq experiments are with three biological replicates. More than 30 million (mean ± standard deviation = 37,392,922 ± 3,100,588) 100 bp paired-end reads for each RNA-Seq sample were generated using an Illumina HiSeq 2000 sequencer in National Center for Genome Medicine, Academia Sinica. The human genome assembly hg38 including un-placed and un-localized scaffolds and RefGene annotation were downloaded from the UCSC Genome Browser on 2017.1.2[48]. Low quality of sequencing bases were trimmed based on the Phred + 33 quality score (>20) from both of the 5′- and 3′-ends of reads. After trimming, reads were discarded, if they were shorter than 75 bp, or had one or more ambiguous base. The alignment, quantification, normalization, and differential expression analysis were performed by STAR 2.5.3a[49] through Partek Flow (Partek Inc.), htseq-count 0.6.0[50], TMM[51], and edgeR 3.18.1[52], respectively. Genes with count-per-million (CPM) values above 1 in at least three samples were retained, and genes with no or low expression levels were discarded. False discovery rate (FDR) < 0.05 was set as a threshold to identify differentially expressed genes. The pathway enrichment analysis using Fisher's exact test was analyzed through the use of IPA (QIAGEN Inc., https://www.qiagenbioinformatics.com/products/ingenuitypathway-analysis). IPA is also used to build a regulatory network for cell differentiation and carcinogenesis regulated by PTEN pathway[53]. In addition, gene set enrichment analysis was performed using GSEA[54]. RNA-seq raw data are accessible at NCBI GEO with accession number GSE105145. All relevant data are available from the authors with restrictions.

**Animals and cell transplantation.** Approximately $10^6$ DPSCs, BMSCs, and BMSCs transduced with control shRNAs or with DNMT3B shRNAs were mixed with 10 mg of hydroxyapatite/tricalcium phosphate (HA/TCP) ceramic powder (Zimmer Inc., Warsaw, IN, USA) and then transplanted subcutaneously into the dorsal surface of 10-week-old NOD/SCID (NOD.CB17-Prkdcscid/NcrCrl, National Laboratory Animal Center). Totally around 10 mice were used. These procedures were performed in accordance with protocols approved by the Animal Care and Use Committee in China Medical University and we have complied with all relevant ethical regulations for animal testing and researches. The transplants were recovered at 8 weeks posttransplantation, fixed with 4% formalin, decalcified with buffered 10% EDTA (pH 8.0), and then embedded in paraffin sections (5 mm) were deparaffinized and stained with H&E and immunohistochemistry of DSP (H-300 sc-33586) (Santa Cruz Biotechnology). For tumorigenesis assay, $10^7$ DPSCs and BMSCs without or with genetic modification were transplanted subcutaneously into 6–10-week-old NOD/SCID mice. Mice were sacrificed for gross and microscopic evaluation of tumor formation 2–3 months later.

**ChIP assay.** To demonstrate the binding of DNMT3B, H3K27me3, and H3K9me2 protein to the PTEN promoter, the ChIP assay was performed using a commercial kit (Millipore) according to the manufacturer's protocol with minor adjustments. The MSCs were grown to confluence, crosslinked with 1% formaldehyde for 8 min at room temperature, washed in ice-cold PBS containing protease inhibitors, then lysed on ice for 10 min in lysis buffer (10 mM Tris HCl, pH 8.0, 1% SDS) with phosphatase and protease inhibitors. DNA–protein complexes were sonicated to 200 and 500 base pairs

(bp). One aliquot of the soluble chromatin was stored at 20 °C for use as input DNA, and the remainder was diluted 10 times in IP buffer (10 mM Tris HCl, pH 8.0, 0.1% SDS, 1% Triton X-100, 1 mM EDTA, and 150 mM NaCl) containing phosphatase and protease inhibitors, and incubated overnight (4 °C) with anti-human DNMT3B (NOVUS Biologicals), H3K27me3 (Millipore), and H3K9me2 (Abcam) antibody. DNA–protein complexes were washed with low LiCl, high LiCl, and TE wash buffer on protein G magnet beads (GE) and then eluted with IP elution buffer. Cross-linking was reversed by incubation at 65 °C for overnight. Proteins were removed with proteinase K, and DNA was extracted with phenol/chloroform, redissolved, and PCR-amplified with specific primer sense, 5′-CTT AGC TCG TTA TCC TCG CCT-3′; antisense, 5′-GAT GAA AGC TGA GAT GGG TGC-3′ for PTEN promoter.

**Statistical analysis.** All values are expressed as mean ± SD. Comparisons between two groups were analyzed by Student t-test. A value of $p < 0.05$ was considered statistically significant.

**Reporting summary.** Further information on research design is available in the Nature Research Reporting Summary linked to this article.

## Data availability
Sequence data that support the findings of this study have been deposited in NCBI GEO server (http://www.ncbi.nlm.nih.gov/geo) with accession numbers as GSE105145 and GSE87624. The authors declare that the data supporting the findings of this study are available within the paper and supplementary information files. Source data for figures are provided with the paper.

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

## Acknowledgements

Grants supported by Minister of Science and Technology (MOST 106-2321-B-039-003-) and Integrative Stem Cell Center, China Medical University Hospital. This work was also financially supported by the "Drug Development Center, China Medical University" from The Featured Areas Research Center Program within the framework of the Higher Education Sprout Project by the Ministry of Education (MOE) in Taiwan. The funding sources had no involvement in study design, in the collection, analysis and interpretation of data, in the writing of the report, and in the decision to submit the article for publication. Some bioinformatics software and computing resources used in the analysis are funded by the University of Southern California (USC) Office of Research and the Norris Medical Library.

## Author contributions

W.-C.S. designed and performed experiments, analyzed the data, and wrote the paper. Y.-C.L., L.-H.L., and K.L. prepared the RNA samples and performed the RNA-seq analysis. H.-C.L. designed and performed experiments. S.-Y.K. and J.W. analyzed the data. C.-M.C. edited the paper. S.-C.H. wrote the proposal, designed and supervised the study, analyzed the data, and wrote the paper.

## Additional information

**Competing interests:** The authors declare no competing interests.

