## [Peer Review File · Nature Communications]

Reviewers' comments:

Reviewer #1 (Remarks to the Author):

This is a well written and well done study that has conducted extensive comparison between BMSCs and DPSCs. They report that DPSCs exhibited increased osteogenic potential, decreased adipogenic potential, formed dentin pulp-like complexes in vivo, and were more resistant to oncogenic transformation as compared to BMSCs. By conducting genome-wide RNA-seq they have identified differences in the profile of gene expression and signaling pathways in the two populations. The most interesting is the discovery that higher PTEN expression in DPSCs than BMSCs was responsible for the lineage commitment and tumorigenesis differences in both cells. Additionally, BMSCs exhibited higher DNA methylation level and repressive mark H3K9Me2 enrichment in the PTEN promoter compared to DPSCs. Overall as commented before, this is a comprehensive study appropriate for publication. The only issues that need to be addressed or commented by the authors is the issues of significant age differences between isolated DPSCs (from primary teeth in 7-10 years old kids) and BMSCs (bone marrow from 27-41 years old individual that undergo orthopedic surgery). The possibility that related differences may or may not be related to age of source of stem cell warrant some discussion.

Reviewer #2 (Remarks to the Author):

In this paper the main question is whether BMSC/DPSC lineage determination and tumorigenesis is regulated by a single factor. More specifically the differences in Osteogenesis and adipogenesis between Dental pulp Stem cells and Bone marrow derived Stem Cells is examined. Using RNA seq and bioinformatics, signaling hubs are identified centering on PTEN. The researchers point out that little is known about PTEN in lineage commitment and tumor formation in DPSC and BMSC. Essentially it is shown that PTEN is expressed higher in DPSC, it is repressed in BMSC by DNMT3B and G9a which induces DNA methylation and H3K9me2 on the PTEN promoter.

This paper reveals novel insights into the mechanisms governing PTEN expression in BMSC and DPSC and provides a mechanism explaining the differences in differentiation potential and potential for tumor formation between the two cell types. The function of PTEN in osteogenesis, adipogenesis and tumorigenesis however is not a novel one. PTEN has been conditionally ablated in mice showing in vivo evidence for its role in bone formation, endochondral ossification, cranial development and adipogenesis for eg (Burgess et al, 2013, PLoSOne., Guntur et al, 2011, Development., Ford Hutchinson et al, 2007, JBMR, Shu Chen et al, 2009, Molecular Carcinogenesis). Its role in Osteosarcoma and tumorigenesis in general has been published yet this paper does shed some light on why DPSC are less likely to form tumors than BMSC.

Questions:

1. Although this paper finds that G9a represses PTEN, which leads to promotion of adipogenesis and inhibition of osteogenesis in BMSC, Deletion of G9a has been shown to increase adipogenesis as G9a methylates the PPAR γ promoter, repressing adipogenesis. (Wang et al, 2013, EMBO J). How do the authors account for this difference ?
2. Are PTEN levels deregulated in human Osteosarcomas?

Specific Remarks:

1. I cannot see difference in colony size for DPSC and BMSC in Figure S1B.
2. Flow cytometric analysis is referred to as Figure S1C in text but is S1B in Figures. S1D in Text is actually S1C in Figures. Also S1E should be S1D?,
3. The Col2A1 staining in figure S1F does not have an antibody control to determine specificity.
4. For Figure 1A, lipid quantitation should be relative to cell number or total DNA content to take proliferation differences of long term cultures into account. This should also apply for all lipid quantitation throughout the text.
5. Alizarin red quantitation for Figure 1C should be relative to DNA or cell number for similar reasons as above.
6. For Figure 1D, should do a time course over time and not just two time points. Runx2 levels go down for DP and BMSC during differentiation? Why? Differences between DP and BMSC could be due to transcriptional kinetic differences and not necessarily absolute levels of gene expression.
7. For Figure S2D there is no oil red positive cells in the control well to say that there is a decrease with the inhibitor. Should use different time point.
8. Magnifications look different for images in Figure 4B. The right image looks higher magnification than the left image.
9. It is vital to quantify figure 6 relative to DNA number or cell number. Knockdown of G9a in BMSC or DPSC can induce rapid cell death, and block proliferation and skew the results.

Reviewer #3 (Remarks to the Author):

The manuscript submitted to Nature Communications entitled "PTEN methylation is associated with lineage commitment and tumorigenesis in adult stem cells" by Prof HUNG and colleagues (reference NCOMMS-18-00244) presents a study and characterization of human adult mesenchymal stem cells (MSCs) isolated from dental pulp (DP) and from bone marrow (BM).

The work is well done and well presented, but I think that the authors should address some important questions and clarify several points before publication.

Major points:

- 1.- A first important issue is that the title is misleading. All the work is done on "mesenchymal stem cells" (MSCs), so this should appear in the title instead of "stem cells". Many articles in the literature (registered in PubMed) use the abbreviation MSC for "mesenchymal stem cells"; so I also strongly recommend changing to this abbreviation (MSC) throughout the entire article. Moreover, the two lineages of MSCs isolated from human dental pulp (DP) and bone marrow (BM) should be named in the same way: DP-MSCs and BM-MSCs.
- 2.- Another important point related to the title, but also to the studies and results presented in the paper is that very little is done in this work about the "tumorigenesis" of the MSCs. In fact, only the last figure (Figure 7) presents some results related to this, and claim a link between the lineage commitment of the DP-MSCs and the tumorigenesis (that apparently is increased when gene PTEN is silenced). The presentation of these results is quite poor and not well shown at all in Figure 7 (that only includes 4 immunohistochemistry microscope images). The formation of an osteosarcoma-like tumor is not well documented and is a topic that is outside of the main argument of the rest of the article, which needs better demonstration. For these reasons I think the "tumorigenesis" should not be present in the title of the work (to avoid misleading) and the results of Figure 7 should be better presented and shown. Finally, according to all these comments, a possible suggestion for a more appropriate title may be: "PTEN activation and methylation are associated with lineage commitment in adult mesenchymal stem cells".

3.- In the introduction of the article and in the discussion there are not many references to other published works that study the same type of human cells, i.e.: adult mesenchymal stem cells (MSCs) isolated from bone marrow (BM) and from other tissues (like adipose tissue, placenta, etc), that also have been done using RNA-seq. In this way, it will be good to include a comment or comparison of the expression signature of BM-MSCs with the expression studies presented in the publications by Roson-Burgo et al. (PubMed PMIDs 25326687 and 27871224). This can be also used as a validation of the genes that appear as most significant and consistent in the expression signature of BM-MSCs (that includes 3064 genes, as indicated in page 9, line 12 of the manuscript). This comparison will also be relevant for the genes included in the PTEN and AKT pathways.

4.- With respect to the analysis and presentation of the results from RNA-seq in Figure 2 several important issues are missing:

(A) The authors say that the "RNA-seq raw data are accessible at NCBI GEO (accession number: GSE105145)", but I could not access such dataset because they do not provide a key or password to really access the data.

(B) Despite placing the data in NCBI GEO they also should provide (at least as a supplementary file) a table with the numbers corresponding the normalized expression signal of the 3064 genes included in Figure 2B, that are selected because they are significant in the contrast of 3 samples of DP-MSCs versus 3 samples of BM-MSCs. The fold-changes and p-values corresponding to the differential expression analysis of these genes should be also included in such table.

(C) After doing the GSEA analysis they select a subset of 272 genes as down-regulated targets of PTEN pathway as a result of the analysis done using IPA Path Explorer. The selection of these gene set should be better proved, because IPA is a commercial bioinformatic program that includes a knowledge-base database that is a "black-box" where nobody can do a checking. So, I recommend that these 272 gene targets of PTEN are also demonstrated using other open-source annotation tool.

(D) Finally, Figure 2E has very little quality and it is not a good representation of the network of 272 PTEN targets. The quality of the figure is very poor. I can not see which genes are up-regulated and which are down-regulated. The red lines look like "links or edges" of the network, but in fact they are used to re-mark 25 PTEN signaling genes, and this is quite confusing. The final conclusion or message from this figure is not clear.

Minor points:

1.- It will be good to include the correct IDs (from NCBI ENTREZ or from ENSEMBL) of the human genes used in this work. I understand that PTEN is "NCBI Gene ID: 5728" and "ENSG00000171862"; but for AKT the gene that the authors use is not so clear (I suppose that it is AKT1, "NCBI Gene ID: 207" and "ENSG00000142208").

Reviewer #1 (Remarks to the Author):

This is a well written and well done study that has conducted extensive comparison between BMSCs and DPSCs. They report that DPSCs exhibited increased osteogenic potential, decreased adipogenic potential, formed dentin pulp-like complexes in vivo, and were more resistant to oncogenic transformation as compared to BMCs. By conducting genome-wide RNA-seq they have identified differences in the profile of gene expression and signaling pathways in the two populations.

The most interesting is the discovery that higher PTEN expression in DPSCs than BMSCs was responsible for the lineage commitment and tumorigenesis differences in both cells. Additionally, BMSCs exhibited higher DNA methylation level and repressive mark H3K9Me2 enrichment in the PTEN promoter compared to DPSCs. Overall as commented before, this is a comprehensive study appropriate for publication. The only issues that need to be addressed or commented by the authors is the issues of significant age differences between isolated DPSCs (from primary teeth in 7-10 years old kids) and BMSCs (bone marrow from 27-41 years old individual that undergo orthopedic surgery).

The possibility that related differences may or may not be related to age of source of stem cell warrant some discussion.

Response:

Thank you for your comment that allows us to greatly improve the quality of our manuscript. To answer this question, we reanalyzed 48 human and rhesus monkey microarray datasets with Affymetrix CEL format, obtained from both NCBI Gene Expression Omnibus (GEO) and EMBL-EBI ArrayExpress repositories (Supplement Data 2). All human mesenchymal stem cells (MSCs) transcriptomes were produced by Affymetrix Human Genome U133 Plus 2.0 Array. Therefore, they can be integrated together to explore the relationship between PTEN and age (Fig. S6A). For the same age range (i.e. from 21 to 40 years old), we found that PTEN levels in dental pulp-derived MSCs are significantly higher than those in bone marrow-derived MSCs (t-test, $t_8 = 5.355$, P-value = 0.0007), consistent with our RNA-Seq results (Fig. S6A). Within human bone marrow-derived MSCs, PTEN is not differentially expressed among different age groups (ANOVA, $F_{2, 31} = 0.494$, P-value = 0.615). For rhesus monkey bone marrow-derived MSCs using Affymetrix Human Genome U133A Array, we also found that PTEN expression levels among different developmental stages are similar (Fig. S6B, ANOVA, $F_{2, 8} = 0.154$, P-value = 0.860).

Furthermore, we also looked into the relationship between PTEN expression and age in other tissue systems in literature. First, Zhao et al. (2017) performed a meta-analysis including 2,486 patients. They found that PTEN expression was significantly correlated with tumor stages and the survival of non-small cell lung cancer patients. However, no association was found between PTEN expression and age. Similarly, there was no significant association between PTEN expression and age when studying early-stage breast cancer (Okabe et al., 2017), colon cancer (Ormanns et al., 2014), pediatric astrocytomas (Rickert, 2004) NK/T-cell lymphomas (Fu et al., 2017), soft tissue sarcomas (Waniczek et al., 2012), and large intestine polyps (Waniczek et al., 2012). The above studies suggest that age likely does not play a major role in our experiments. Finally, we thank you for this kind suggestion again.

We also discuss this in the discussion section of the revised version. “There are still limitations in the current study. Because the ages of the donors who provided the samples were not consistent between isolated DP-MSCs and BM-MSCs, the related differences may be related to the age of source of stem cells. To deal with this issue, we integrated 48 human and rhesus monkey microarray datasets downloaded from public repositories (Supplementary Data 2). Both human and rhesus monkey expression profiles manifested that PTEN expression levels in BM-MSCs are similar among different ages and developmental stages (Fig. S6). By contrast, PTEN is significantly up-regulated in DP-MSCs compared to BM-MSCs in the same age donors, consistent with our findings (Fig. S6A). These integrated analyses have revealed that age is not a major confounding factor in this study and further reinforced our discovery.” from page 19, line 16 to page 20, line 7.

Reviewer #2 (Remarks to the Author):

In this paper the main question is whether BMSC/DPSC lineage determination and tumorigenesis is regulated by a single factor. More specifically the differences in Osteogenesis and adipogenesis between Dental pulp Stem cells and Bone marrow derived Stem Cells is examined. Using RNA seq and bioinformatics, signaling hubs are identified centering on PTEN. The researchers point out that little is known about PTEN in lineage commitment and tumor formation in DPSC and BMSC. Essentially it is shown that PTEN is expressed higher in DPSC, it is repressed in BMSC by DNMT3B and G9a which induces DNA methylation and H3K9me2 on the PTEN promoter.

This paper reveals novel insights into the mechanisms governing PTEN expression in BMSC and DPSC and provides a mechanism explaining the differences in differentiation potential and potential for tumor formation between the two cell types. The function of PTEN in osteogenesis, adipogenesis and tumorigenesis however is not a

novel one. PTEN has been conditionally ablated in mice showing in vivo evidence for its role in bone formation, endochondral ossification, cranial development and adipogenesis for eg (Burgess et al, 2013, PLoSOne., Guntur et al, 2011, Development., Ford Hutchinson et al, 2007, JBMR, Shu Chen et al, 2009, Molecular Carcinogenesis). Its role in Osteosarcoma and tumorigenesis in general has been published yet this paper does shed some light on why DPSC are less likely to form tumors than BMSC.

Question 1:

1. Although this paper finds that G9a represses PTEN, which leads to promotion of adipogenesis and inhibition of osteogenesis in BMSC, deletion of G9a has been shown to increase adipogenesis as G9a methylates the PPAR γ promoter, repressing adipogenesis. (Wang et al., 2013, EMBO J). How do the authors account for this difference ?

Response 1:

Thank you for your kind note. In this paper, Wang et al showed that (1) high levels of H3K9me2 were found on the master adipogenic transcription factor PPAR γ in undifferentiated 3T3-L1 (Wang et al., 2013). In contrast, H3K9me2 levels were low on gene loci encoding other adipogenic transcription factors, including C/EBP α , C/EBP β , C/EBP δ , KLF4, Krox20 and CREB. (2) Consistent with previous reports (Cho et al, 2009; Mikkelsen et al, 2010; Wang et al, 2010), H3K4me3 levels on PPAR γ 1 and γ 2 promoters increased after differentiation of 3T3-L1 white preadipocytes. Interestingly, H3K9me2 levels on the entire PPAR γ locus decreased markedly after differentiation. Similar results were obtained during adipogenesis of brown adipocytes. The authors further demonstrate that G9a protein level was much lower in adipocyte than in preadipocytes of white adipose tissue. However, the mechanism that mediates G9a downregulation from preadipocyte to adipocyte in the defined medium (0.5mM IBMX, 2mg/ml dexamethasone, and 0.125mM indomethacin) has not been explored. (3) The authors further used SV40T-immortalized G9a $^{-/-}$ preadipocyte to study the role of G9a in repressing adipogenesis. However, two of the three immortalized G9a $^{-/-}$ brown preadipocyte cell lines (#2 and #3) showed full adipogenesis potential under the standard induction condition. Based on these data, enrichment patterns of H3K9me2 on gene loci encoding adipogenic transcription factors, such as PPAR γ and other adipogenic transcription factors, including C/EBP α , C/EBP β , C/EBP δ , KLF4, Krox20 and CREB are different in a transcription factor-dependent manner. Moreover, both of the enrichment of H3K9me2 on PPAR γ locus and G9a level changed along with the differentiation status. These data suggest both of H3K9me2 enrichment and G9a level are dynamic before and after adipogenesis, suggesting a context-dependent manner. Consistent with this possibility, a recent paper reported that knockdown of H3K4/K9

demethylase LSD1 in 3T3-L1 preadipocytes leads to increased H3K9me2 and decreased H3K4 me2 on C/EBP α promoter, decreased C/EBP α expression in preadipocytes and impaired adipogenesis (Musri et al., 2010), although it was unclear whether LSD1 regulates H3K9me2 on PPAR γ promoter.

Our cell source is human BM-MSC. It is quite different from Wang's murine primary brown preadipocytes, murine primary white preadipocytes and the immortalized murine white preadipocyte cell line 3T3-L1. Therefore, H3K9me2 enrichment on either PPAR γ or PTEN loci and G9a level may also differ in BM-MSC in comparison with murine preadipocyte lines. Therefore, these differences in BM-MSCs and 3T3-L1 may attribute to the different roles of H3K9me2 enrichment and G9a in adipogenesis.

Because Wang's paper is regarding the roles of H3K9me2 enrichment and G9a in adipogenesis of murine preadipocytes, which is quite different from adipogenesis of human BM-MSC, we suggest not including the discussion of their data to our manuscript.

Question 2:

2. Are PTEN levels deregulated in human Osteosarcomas?

Response 2:

Thank you very much for the comment. We believe that PTEN levels are deregulated in human osteosarcomas (OS). Freeman et al. (2008) summarized that loss of PTEN was common in OS. More importantly, we re-analyzed human RNA-Seq data from Gene Expression Omnibus (GEO) accession GSE87624 (Scott et al., 2018), including 3 normal bone samples, 44 patient OS samples, and 5 OS cell lines. We found that PTEN levels are significantly different among the three groups (Fig. S5, ANOVA, $F_{2,49} = 9.576$, P-value = 0.0003). Using normal bone samples as the control, PTEN levels are significantly down-regulated in human patient OS samples (Tukey's HSD post-hoc test, P-value = 0.0196). PTEN levels in human OS cell lines are significantly down-regulated compared to normal bones (Tukey's HSD post-hoc test, P-value = 0.0002). Consistently, Xi and Chen (2017) found that PTEN is reduced in human OS cell lines as compared to normal human osteoblasts. Gong et al. (2017) found that PTEN is expressed at low level in human osteosarcomas compared to adjacent tissues. Moreover, the down-regulation of PTEN enhanced migration and invasion of osteosarcoma cell lines (Tian et al., 2015) and is correlated with poor prognosis in human osteosarcomas (Li et al., 2016; Tian et al., 2014). In short, these studies consistently demonstrated that PTEN levels are down-regulated in human osteosarcomas. These new figures are added to the manuscript as Figure S5.

We also added these data at the end of the result section. “To support our findings, we analyzed human RNA-Seq data of 3 normal bone samples, 44 osteosarcomas (OS) patient samples, and 5 OS cell lines from Gene Expression Omnibus (GEO) accession GSE87624 (Scott et al., 2018). We found that PTEN levels are significantly different among the three groups (Fig. S5, ANOVA, $F_{2,49} = 9.576$, P-value = 0.0003). Using normal bone samples as the control, PTEN levels are significantly down-regulated in human patient OS samples (Tukey HSD post-hoc test, P-value = 0.0196). PTEN levels in human OS cell lines are more significantly down-regulated compared to normal bones (Tukey HSD post-hoc test, P-value = 0.0002). These comparisons support that PTEN levels were deregulated in human osteosarcomas.” on page 15, line 6-16.

Specific Remarks:

1. I cannot see difference in colony size for DPSC and BMSC in Figure S1B.

Response:

Thank you very much for this excellent observation. In the manuscript, we clarified that we mean the cell size. We further made a partial enlargement in Fig. S1B, and we can clearly see the difference.

2. Flow cytometric analysis is referred to as Figure S1C in text but is S1B in Figures. S1D in Text is actually S1C in Figures. Also S1E should be S1D?

Response:

We thank you for pointing out these mistakes and we apologize. The appropriate corrections have been made.

3. The Col2A1 staining in figure S1F does not have an antibody control to determine specificity.

Response:

Thank you for your kind note. We have now added a negative control without the 1st antibody in figure S1F in the revised version.

4. For Figure 1A, lipid quantitation should be relative to cell number or total DNA content to take proliferation differences of long term cultures into account. This should also apply for all lipid quantitation throughout the text.

Response:

We would like to thank you for the valuable comments. For *in vitro* differentiation into adipocytes, cells were induced with adipogenic induction medium (AIM). After the appearance of morphologic features of adipogenic lineage, cells in AIM were stained with Oil Red O solution. For quantitative analysis, the stained lipid droplets were dissolved in 100% isopropanol, followed by quantitation of the Oil Red O extract by measuring the O.D. at 510 nm. Because the cells had been fixed with formalin, stained with Oil Red O, and extracted with isopropanol, which induces cell membrane rupture and the loss of DNA content, they were not suitable for requantitation of DNA (below Figure A). Moreover, the cell numbers can not be appropriately calculated after differentiation, especially at late differentiation when tissues have been formed. Thus, we are not able to perform quantitation of lineage differentiation relative to cell number or total DNA content. This limitation is also applied to Alizarin red staining for osteogenic differentiation.

To prevent the contribution of proliferation differences to the differences in lineage differentiation, we actually seeded cells at a high cell density ($10^4/\text{cm}^2$, refer to M&M page 22) in complete growth medium and replaced with defined differentiation medium overnight or 24 h later when cells became almost of complete confluence. When cells initiated differentiation, they would not proliferate. We also cultured a panel of cells for DNA quantitation at different periods of differentiation, and found there was no significant difference in total DNA content at 1, 2 and 3 weeks of adipogenic differentiation and osteogenic differentiation between DPSCs and BMSCs (below Figure B, C). These data suggest that the differences in adipogenic and osteogenic differentiation between DPSCs and BMSCs were not contributed by proliferation differences between these two cells.

In addition, we also suggest the reviewer refer to the published papers (El Agha et al., 2017; Kramann et al., 2016; Song et al., 2016). The authors did not perform lipid or mineral quantitation relative to cell number or total DNA content. We believe they also encountered the same problems with us when performing the quantitation.

5. Alizarin red quantitation for Figure 1C should be relative to DNA or cell number for similar reasons as above.

Response:

For *in vitro* differentiation into osteoblasts, cells were induced with osteogenic induction medium (OIM). After 3 weeks in OIM, cells were stained by Alizarin Red S. Since the cells had been fixed with ice-cold 70% ethanol and extracted with cetylpyridinium chloride (CPC) buffer, they were not suitable for requantification of DNA (below Figure A). Therefore, similar to the above, we cultured a batch of cells under the same conditions and quantified DNA without staining. We found there was no difference in total DNA between the two cell types at 7, 14, or 21 days (below Figure B, C). Therefore, the data suggest that the higher osteogenic potential observed in DPSCs over BMSCs is not because of differences in the cell number.

In addition, we refer to the publish paper (Ambrosi et al., 2017; Guimaraes-Camboa et al., 2017; Pillai et al., 2017) they did not perform quantitation relative to cell number or total DNA content to take proliferation differences of long term cultures into account.

6. For Figure 1D, should do a time course over time and not just two time points. Runx2 levels go down for DP and BMSC during differentiation? Why? Differences between

DP and BMSC could be due to transcriptional kinetic differences and not necessarily absolute levels of gene expression.

Response:

Response: Thank you for your great comments. Expression of transcription factors or lineage genes over the duration of differentiation is known to be dynamic (Stein et al., 2004). For example, Runx2, a transcription factor of osteogenesis is only expressed at early stages of differentiation, while osteocalcin (OC) – representing mineralization – is only expressed at late stages of differentiation (below Figure) (Perinpanayagam et al., 2006).

To address potential differences in transcriptional kinetics, we repeated the experiment to include a Day 7 time point; the new experiments were repeated three times. The new data confirms that Runx2 was expressed at high levels at early stages (Day 3) in DP-MSCs and was attenuated at later stages (Day 7). There was a slight shift in the timeline compared to the previous data, but which is likely due to experimental variation. In BM-MSCs, Runx2 was expressed throughout the duration

of differentiation but was not expressed at such high levels as observed in DP-MSCs. We have also revised the Figure 1D in the revised version.

7. For Figure S2D there is no oil red positive cells in the control well to say that there is a decrease with the inhibitor. Should use different time point.

Response:

Thank you for the kind suggestion. We have revised Figure S2D to include a new panel of images showing stronger Oil Red oil staining of the BM-MSCs at Day 14. These images better represent the quantified Oil Red oil staining in Figure S2D of the revised version.

8. Magnifications look different for images in Figure 4B. The right image looks higher magnification than the left image.

Response:

Thank you for this excellent observation. We have confirmed that the magnification of this image is correct. The PTEN O/E cells appear larger because they are more spread; the size of the nucleus in both images are similar.

9. It is vital to quantify figure 6 relative to DNA number or cell number. Knockdown of G9a in BMSC or DPSC can induce rapid cell death, and block proliferation and skew the results.

Response:

Thank you very much for the great suggestion. We agree with your comment that G9a knockdown in BM-MSCs did reduce cell growth rate or induce senescence probably through upregulating PTEN to inactivate AKT (Fig. 6E), the pathway important for cell proliferation (Sheng et al., 2017). We have been struggling to get enough cell numbers ($10^4/\text{cm}^2$, refer to M&M page 22) for the initiation of adipogenesis (Fig. 6F) and osteogenesis (Fig. 6G). Despite the difficulties, we have repeated these experiments for many times. We found that G9a knockdown inhibited adipogenesis, while promoting osteogenesis. The answer of this question is also the same as we answered your question 4 and 5. We induced differentiation when cells became confluent, which prevented the contribution of cell growth effect (either proliferation or death) to differentiation. We also excluded the possibility of cell death during differentiation. If the cells underwent cell death or apoptosis, it was impossible to have an increased osteogenesis (Fig. 6G). To improve the quality of Figure 6, we have also revised the Figure 6D by replacing it with more beautiful pictures and have further quantified the data in the revised version.

Reviewer #3 (Remarks to the Author):

The manuscript submitted to Nature Communications entitled "PTEN methylation is associated with lineage commitment and tumorigenesis in adult stem cells" by Prof HUNG and colleagues (reference NCOMMS-18-00244) presents a study and characterization of human adult mesenchymal stem cells (MSCs) isolated from dental pulp (DP) and from bone marrow (BM).

The work is well done and well presented, but I think that the authors should address some important questions and clarify several points before publication.

Major points:

1.- A first important issue is that the title is misleading. All the work is done on "mesenchymal stem cells" (MSCs), so this should appear in the title instead of "stem cells". Many articles in the literature (registered in PudMed) use the abbreviation MSC for "mesenchymal stem cells"; so I also strongly recommend changing to this abbreviation (MSC) throughout the entire article. Moreover, the two lineages of MSCs isolated from human dental pulp (DP) and bone marrow (BM) should be named in the same way: DP-MSCs and BM-MSCs.

Response:

The suggested corrections have been made. The title now includes the phrase "mesenchymal stem cells". We have changed the abbreviation of "mesenchymal stem cells" to MSCs throughout the entire article. Moreover, the two lineages of MSCs isolated from human dental pulp (DP) and bone marrow (BM) were also changed to DP-MSCs and BM-MSCs, respectively.

2.- Another important point related to the title, but also to the studies and results presented in the paper is that very little is done in this work about the "tumorigenesis" of the MSCs. In fact, only the last figure (Figure 7) presents some results related to this, and claim a link between the lineage commitment of the DP-MSCs and the tumorigenesis (that apparently is increased when gene PTEN is silenced). The presentation of these results is quite poor and not well shown at all in Figure 7 (that only includes 4 immunohistochemistry microscope images). The formation of an osteosarcoma-like tumor is not well documented and is a topic that is outside of the main argument of the rest of the article, which needs better demonstration. For these reasons I think the "tumorigenesis" should not be present in the title of the work (to avoid misleading) and the results of Figure 7 should be better presented and shown.

Finally, according to all these comments, a possible suggestion for a more appropriate title may be: "PTEN activation and methylation are associated with lineage commitment in adult mesenchymal stem cells".

Response:

Thank you very much for the great comments. We have now added a figure panel to describe the process of the experiment, which makes it easier to be understood (Fig. 7A). "To support our findings, we analyzed human RNA-Seq data of 3 normal bone samples, 44 osteosarcomas patient samples, and 5 osteosarcomas cell lines from Gene Expression Omnibus (GEO) accession GSE87624(Scott et al., 2018). We found that PTEN levels are significantly different among the three groups (Fig. S5, ANOVA, $F_{2,49} = 9.576$, P-value = 0.0003). Using normal bone samples as the control, PTEN levels are significantly down-regulated in human patient osteosarcomas samples (Tukey HSD post-hoc test, P-value = 0.0196). PTEN levels in human osteosarcoma cell lines are more significantly down-regulated compared to normal bones (Tukey HSD post-hoc test, P-value = 0.0002). These comparisons support that PTEN levels were deregulated in human osteosarcomas." This part was added on Page 15, Line 6-16.

We have also modified the title after considering the comment by the Reviewer: "PTEN activation and methylation are associated with lineage commitment and imply tumorigenesis in mesenchymal stem cell". Instead of using the term "are associated with", we replaced it with "imply". We hope the reviewer can agree to keep "tumorigenesis" in the title.

3.- In the introduction of the article and in the discussion there are not many references to other published works that study the same type of human cells, i.e.: adult mesenchymal stem cells (MSCs) isolated from bone marrow (BM) and from other tissues (like adipose tissue, placenta, etc), that also have been done using RNA-seq. In this way, it will be good to include a comment or comparison of the expression signature of BM-MSCs with the expression studies presented in the publications by Roson-Burgo et al. (PubMed PMIDs 25326687 and 27871224). This can be also used as a validation of the genes that appear as most significant and consistent in the expression signature of BM-MSCs (that includes 3064 genes, as indicated in page 9, line 12 of the manuscript). This comparison will also be relevant for the genes included in the PTEN and AKT pathways.

Response:

We appreciate this great comment. Comparing published works makes the novelty of this study stand out more. In the mentioned papers, PTEN is not a functional signature of BM-MSCs when compared to placenta (PL)-MSCs (Roson-Burgo et al., 2014), adipose (AD)-MSCs, hematopoietic stem and progenitor cells (HSPCs), skin

fibroblasts, and osteoblasts (Roson-Burgo et al., 2016). On the other hand, in our study, PTEN is a functional signature of BM-MSCs. There are at least two non-mutual exclusive reasons to interpret the novel signature of BM-MSCs that we discovered. Firstly, the comparative tissue in our study (i.e. dental pulp (DP)-MSCs) is an extremely rare tissue type with a very low tumorigenic potential (Neuhaus, 2007). Therefore, highly contrasting tissues of tumorigenic abilities will reveal the signatures of differential tumorigenesis more conspicuously. In addition, the key regulator for lineage commitment in DP-MSCs may be different from PL-MSCs, AD-MSCs, HSPCs, fibroblasts, and osteoblasts. Consequently, in using DP-MSCs as a comparative tissue, we have identified novel functional signatures of BM-MSCs for both lineage commitment and tumorigenesis.

We also addressed these results in the introduction of the revised version: “RNA deep sequencing (RNA-Seq) analysis of transcriptome profiles revealed the differences between BM-MSC and placenta MSC; the former increases in genes enrolled in micro-environmental process, such as bone formation and blood vessel morphogenesis, while the latter increases in genes associated with mitosis and embryonic morphogenesis (Roson-Burgo et al., 2014). Besides the differences between different tissue origins, the transcriptome profiles also differ in MSC of different commitment states (Roson-Burgo et al., 2016).” from page 4, line 16 to page 5, line 3.

We also discussed these data in the discussion section of the revised version:

“However, PTEN is not a molecular or functional signature of BM-MSCs when compared to placenta MSCs, adipose MSCs, hematopoietic stem and progenitor cells, skin fibroblasts, or osteoblasts (Roson-Burgo et al., 2014). We believe that the characteristics of DP-MSCs, including extremely low tumorigenic potential (Neuhaus, 2007) and unique cell fate, are critical for the development of novel functional signatures between DP-MSCs and BM-MSCs.” on page 17, line 7-12.

4.- With respect to the analysis and presentation of the results from RNA-seq in Figure 2 several important issues are missing:

(A) The authors say that the "RNA-seq raw data are accessible at NCBI GEO (accession number: GSE105145)", but I could not access such dataset because they do not provide a key or password to really access the data.

Response: Please use the token (i.e. obkxggwebrsnvgf) to access our private RNA-Seq data on GSE105145.

(B) Despite placing the data in NCBI GEO they also should provide (at least as a supplementary file) a table with the numbers corresponding the normalized expression

signal of the 3064 genes included in Figure 2B, that are selected because they are significant in the contrast of 3 samples of DP-MSCs versus 3 samples of BM-MSCs. The fold-changes and p-values corresponding to the differential expression analysis of these genes should be also included in such table.

Response: We have now provided supplementary data 1 with gene expression level, fold-change, p-value, and false discovery rate for the 3064 genes.

(C) After doing the GSEA analysis they select a subset of 272 genes as down-regulated targets of PTEN pathway as a result of the analysis done using IPA Path Explorer. The selection of these gene set should be better proved, because IPA is a commercial bioinformatic program that includes a knowledge-base database that is a "black-box" where nobody can do a checking. So, I recommend that these 272 gene targets of PTEN are also demonstrated using other open-source annotation tool.

Response: Each piece of biological data (or finding) in IPA can be confirmed by accessing the original source (article/abstract/third party database) where the relationship is derived from. Using a freely available software STRING (Franceschini et al., 2013; Szklarczyk et al., 2011; Szklarczyk et al., 2015), we further analyzed the gene network, including PTEN pathway and 272 candidate downstream targets. We find that these genes formed a highly interconnected network, and PTEN seemed to be in the central hub (below Figure). There were 1,283 expected interactions in the STRING database. However, we found 2,514 possible protein-protein interactions (PPI enrichment p-value = 1.0×10^{-16}). IPA and STRING are independently pathway analysis tools based on independently databases (Muller et al., 2011). These results further suggest that PTEN and its candidate downstream genes form a tightly linked module to orchestrate the abilities of adipocyte and bone cell differentiation and bone marrow neoplasm for mesenchymal stem cells (MSCs).

We have included a new figure Supplement Figure X1 to describe these results.

PTEN pathway and candidate downstream genes form a tightly interconnected module. The edges are protein-protein interactions in the STRING database. Thicker lines indicate a higher strength of data supporting these interactions. The color used for nodes has no particular meaning.

(D) Finally, Figure 2E has very little quality and it is not a good representation of the network of 272 PTEN targets. The quality of the figure is very poor. I can not see which genes are up-regulated and which are down-regulated. The red lines look like "links or edges" of the network, but in fact they are used to re-mark 25 PTEN signaling genes, and this is quite confusing. The final conclusion or message from this figure is not clear.

Response: We appreciate this constructive comment and acknowledge that it is not easy to show more than 200 genes in a sub-figure. In the revised high-resolution figure, confusing red lines have been removed, and 25 PTEN signaling genes are highlighted by the blue color and moved to the central region. By contrast, adipocyte and bone cell differentiation genes and bone marrow neoplasm genes are moved to the peripheral region. With larger font size and brighter colors, we believe that readers should see which genes are upregulated and which are downregulated.

Minor points:

- 1.- It will be good to include the correct IDs (from NCBI ENTREZ or from ENSEMBL)

of the human genes used in this work. I understand that PTEN is "NCBI Gene ID: 5728" and "ENSG00000171862"; but for AKT the gene that the authors use is not so clear (I suppose that it is AKT1, "NCBI Gene ID: 207" and "ENSG00000142208").

Response:

For RNA-Seq analysis, the gene symbols and corresponding NCBI RefSeq accession numbers (transcript ids) are: PTEN (NM_001304718, NM_001304717), AKT1 (NM_005163, NM_001014432, NM_001014431), AKT2 (NM_001243028, NM_001626, NM_001243027, NM_001330511), and AKT3 (NM_181690, NM_181690_2, NM_005465, NM_005465_2, NM_001206729, NM_001206729_2). We quantify gene expression levels and ignore alternative splicing. Moreover, the Cell Signaling antibody against Phospho-AKT (9271) we used cannot distinguish AKT1, AKT2, and AKT3.

Reference

- Ambrosi, T.H., Scialdone, A., Graja, A., Gohlke, S., Jank, A.M., Bocian, C., Woelk, L., Fan, H., Logan, D.W., Schurmann, A., *et al.* (2017). Adipocyte accumulation in the bone marrow during obesity and aging impairs stem cell-based hematopoietic and bone regeneration. *Cell Stem Cell* *20*, 771-+.
- El Agha, E., Moiseenko, A., Kheirollahi, V., De Langhe, S., Crnkovic, S., Kwapiszewska, G., Kosanovic, D., Schwind, F., Schermuly, R.T., Henneke, I., *et al.* (2017). Two-way conversion between lipogenic and myogenic fibroblastic phenotypes marks the progression and resolution of lung fibrosis. *Cell Stem Cell* *20*, 261-+.
- Franceschini, A., Szklarczyk, D., Frankild, S., Kuhn, M., Simonovic, M., Roth, A., Lin, J.Y., Minguetz, P., Bork, P., von Mering, C., *et al.* (2013). STRING v9.1: protein-protein interaction networks, with increased coverage and integration. *Nucleic Acids Res* *41*, D808-D815.
- Freeman, S.S., Allen, S.W., Ganti, R., Wu, J.R., Ma, J., Su, X.P., Neale, G., Dome, J.S., Daw, N.C., and Khoury, J.D. (2008). Copy number gains in EGFR and copy number losses in PTEN are common events in osteosarcoma tumors. *Cancer* *113*, 1453-1461.
- Fu, X.R., Zhang, X.D., Gao, J.L., Li, X., Zhang, L., Li, L., Wang, X.H., Sun, Z.C., Li, Z.M., Chang, Y., *et al.* (2017). Phosphatase and tensin homolog (PTEN) is down-regulated in human NK/T-cell lymphoma and corrects with clinical outcomes. *Medicine (Baltimore)* *96*, 6.
- Gong, T., Su, X.T., Xia, Q., Wang, J.G., and Kan, S.L. (2017). Expression of NF-kappa B and PTEN in osteosarcoma and its clinical significance. *Oncol Lett* *14*, 6744-6748.
- Guimaraes-Camboa, N., Cattaneo, P., Sun, Y.F., Moore-Morris, T., Gu, Y., Dalton, N.D., Rockenstein, E., Maslah, E., Peterson, K.L., Stallcup, W.B., *et al.* (2017). Pericytes of multiple organs do not behave as mesenchymal stem cells in vivo. *Cell Stem Cell* *20*,

345-+.

Kramann, R., Goettsch, C., Wongboonsin, J., Iwata, H., Schneider, R.K., Kuppe, C., Kaesler, N., Chang-Panesso, M., Machado, F.G., Gratwohl, S., *et al.* (2016). Adventitial MSC-like cells are progenitors of vascular smooth muscle cells and drive vascular calcification in chronic kidney disease. *Cell Stem Cell* *19*, 628-642.

Li, S., Gao, Y., Wang, Y., Wang, K., Dai, Z.P., Xu, D., Liu, W., Li, Z.L., Zhang, Z.D., Yang, S.H., *et al.* (2016). Serum microRNA-17 functions as a prognostic biomarker in osteosarcoma. *Oncol Lett* *12*, 4905-4910.

Muller, T., Schrotter, A., Loosse, C., Helling, S., Stephan, C., Ahrens, M., Uszkoreit, J., Eisenacher, M., Meyer, H.E., and Marcus, K. (2011). Sense and nonsense of pathway analysis software in proteomics. *J Proteome Res* *10*, 5398-5408.

Musri, M.M., Carmona, M.C., Hanzu, F.A., Kaliman, P., Gomis, R., and Parrizas, M. (2010). Histone demethylase LSD1 regulates adipogenesis. *J Biol Chem* *285*, 30034-30041.

Neuhaus, K.W. (2007). Teeth: malignant neoplasms in the dental pulp? *The Lancet Oncology* *8*, 75-78.

Okabe, M., Toh, U., Iwakuma, N., Saku, S., Akashi, M., Kimitsuki, Y., Seki, N., Kawahara, A., Ogo, E., Itoh, K., *et al.* (2017). Predictive factors of the tumor immunological microenvironment for long-term follow-up in early stage breast cancer. *Cancer Sci* *108*, 81-90.

Ormanns, S., Neumann, J., Horst, D., Kirchner, T., and Jung, A. (2014). WNT signaling and distant metastasis in colon cancer through transcriptional activity of nuclear beta-Catenin depend on active PI3K signaling. *Oncotarget* *5*, 2999-3011.

Perinpanayagam, H., Martin, T., Mithal, V., Dahman, M., Marzec, N., Lampasso, J., and Dziak, R. (2006). Alveolar bone osteoblast differentiation and Runx2/Cbfa1 expression. *Arch Oral Biol* *51*, 406-415.

Pillai, I.C.L., Li, S., Romay, M., Lam, L., Lu, Y., Huang, J., Dillard, N., Zemanova, M., Rubbi, L., Wang, Y.B., *et al.* (2017). Cardiac fibroblasts adopt osteogenic fates and can be targeted to attenuate pathological heart calcification. *Cell Stem Cell* *20*, 218-+.

Rickert, C.H. (2004). Prognosis-related molecular markers in pediatric central nervous system tumors. *J Neuropathol Exp Neurol* *63*, 1211-1224.

Roson-Burgo, B., Sanchez-Guijo, F., Del Canizo, C., and De Las Rivas, J. (2014). Transcriptomic portrait of human Mesenchymal Stromal/Stem cells isolated from bone marrow and placenta. *BMC Genomics* *15*, 18.

Roson-Burgo, B., Sanchez-Guijo, F., Del Canizo, C., and Rivas, J.D. (2016). Insights into the human mesenchymal stromal/stem cell identity through integrative transcriptomic profiling. *BMC Genomics* *17*, 27.

Scott, M.C., Temiz, N.A., Sarver, A.E., LaRue, R.S., Rathe, S.K., Varshney, J., Wolf, N.K.,

Moriarity, B.S., O'Brien, T.D., Spector, L.G., *et al.* (2018). Comparative transcriptome analysis quantifies immune cell transcript levels, metastatic progression, and survival in osteosarcoma. *Cancer Res* 78, 326-337.

Sheng, L.L., Mao, X.Y., Yu, Q.X., and Yu, D. (2017). Effect of the PI3K/AKT signaling pathway on hypoxia-induced proliferation and differentiation of bone marrow-derived mesenchymal stem cells. *Exp Ther Med* 13, 55-62.

Song, G.Q., Pacher, M., Balakrishnan, A., Yuan, Q.G., Tsay, H.C., Yang, D.K., Reetz, J., Brandes, S., Dai, Z., Putzer, B.M., *et al.* (2016). Direct reprogramming of hepatic myofibroblasts into hepatocytes in vivo attenuates liver fibrosis. *Cell Stem Cell* 18, 797-808.

Stein, G.S., Lian, J.B., van Wijnen, A.J., Stein, J.L., Montecino, M., Javed, A., Zaidi, S.K., Young, D.W., Choi, J.Y., and Pockwinse, S.M. (2004). Runx2 control of organization, assembly and activity of the regulatory machinery for skeletal gene expression. *Oncogene* 23, 4315-4329.

Szklarczyk, D., Franceschini, A., Kuhn, M., Simonovic, M., Roth, A., Minguéz, P., Doerks, T., Stark, M., Muller, J., Bork, P., *et al.* (2011). The STRING database in 2011: functional interaction networks of proteins, globally integrated and scored. *Nucleic Acids Res* 39, D561-D568.

Szklarczyk, D., Franceschini, A., Wyder, S., Forslund, K., Heller, D., Huerta-Cepas, J., Simonovic, M., Roth, A., Santos, A., Tsafou, K.P., *et al.* (2015). STRING v10: protein-protein interaction networks, integrated over the tree of life. *Nucleic Acids Res* 43, D447-D452.

Tian, K., Di, R., and Wang, L. (2015). MicroRNA-23a enhances migration and invasion through PTEN in osteosarcoma. *Cancer Gene Ther* 22, 351-359.

Tian, Z., Guo, B., Yu, M., Wang, C., Zhang, H., Liang, Q.F., Jiang, K.L., and Cao, L. (2014). Upregulation of micro-ribonucleic acid-128 cooperating with downregulation of PTEN confers metastatic potential and unfavorable prognosis in patients with primary osteosarcoma. *OncoTargets Ther* 7, 1601-1608.

Wang, L., Xu, S., Lee, J.E., Baldridge, A., Grullon, S., Peng, W., and Ge, K. (2013). Histone H3K9 methyltransferase G9a represses PPARgamma expression and adipogenesis. *EMBO J* 32, 45-59.

Waniczek, D., Snietura, M., Piglowski, W., Rdes, J., Kopec, A., Mlynarczyk-Liszka, J., Rudzki, M., Hudyka, K., Arendt, J., and Lange, D. (2012). Analysis of PTEN expression in large intestine polyps and its relation to the recognized histopathological and clinical risk factors for cancer development in this location. *Wspolczesna Onkol* 16, 310-315.

Xi, Y.M., and Chen, Y. (2017). PTEN plays dual roles as a tumor suppressor in osteosarcoma cells. *J Cell Biochem* 118, 2684-2692.

Zhao, Y.S., Zheng, R.Y., Li, J., Lin, F., and Liu, L.X. (2017). Loss of phosphatase and tensin homolog expression correlates with clinicopathological features of non-small cell lung cancer patients and its impact on survival: A systematic review and meta-analysis. *Thorac Cancer* 8, 203-213.

Reviewers' comments:

Reviewer #2 (Remarks to the Author):

All my questions were answered appropriately. I still have some reservations about quantifying mineral production and lipid. This I believe is vital when manipulating genes like G9a, PTEN etc that can induce cell death or change proliferation and can skew results. One can use Arsenazo III kit to measure solubilized mineral. DNA can be extracted from the fixed cells that remain behind using PBS/proteinase K and then using picogreen to measure DNA content. For lipid, can use a combination of Nile red staining and DAPI to count lipid producing cells and total cells in field of view when counting DAPI stained cells.

I do believe quantitating mineral and lipid would be beneficial and can be done. Especially if it is to be published in this journal. I think this is vital for experiments concerning G9a. Knocking down or deletion of G9a drastically affects proliferation and promotes PCD. This skews the results when visually looking at mineral or fat. Therefore it has to be quantified and made relative to cell number or DNA content per well. Once cells form required mineral, researchers can add 0.6M HCl to the well for 2 hours. Then measure the dissolved mineral in the HCl using the mineral stain Calcium Arsenazo111 from Pointe Scientific. The cells that remain behind in the well can be treated with PBS/proteinase at 56 degrees for one hour and DNA quantitated using Pico green. I routinely do this on BMSC, DPSC, adipose derived stem cells.

Lipid can be easily stained with Nile red for 15 minutes and then counterstained with DAPI. Pictures can be taken on fluorescent microscope for Nile red positive cells and DAPI stained cells. Nile red positive/DAPI positive multiplied by 100 would give percentage.

They can do the above for the most critical experiment or maybe show that when G9a or PTEN is manipulated in their hands, there is not a dramatic change in cell number?

As for reviewer 1 comments in regards to age of cells, it is a valid point however most experiments are examining treated versus control within the same population of cells so it becomes somewhat a redundant point. I think it is adequately addressed."

Reviewer #3 (Remarks to the Author):

About the manuscript NCOMMS-18-00244 submitted to Nature Communications, entitled "PTEN activation and methylation are associated with lineage commitment and imply tumorigenesis in mesenchymal stem cell" by Prof HUNG and colleagues (previous title: "PTEN methylation is associated with lineage commitment and tumorigenesis in adult stem cells"), that presents a study and characterization of human adult mesenchymal stem cells (MSCs) isolated from dental pulp (DP) and from bone marrow (BM): I reviewed the responses my questions and the corrections introduced in the manuscript and, in my opinion, the article has greatly improved and can now be accepted for publication.

Point-by-point responses to comments:

Reviewer #2 (Remarks to the Author):

All my questions were answered appropriately. I still have some reservations about quantifying mineral production and lipid. This i believe is vital when manipulating genes like G9a, PTEN etc that can induce cell death or change proliferation and can skew results. **One can use Arsenazo III kit to measure solubilized mineral. DNA can be extracted from the fixed cells that remain behind using PBS/proteinase K and then using picogreen to measure DNA content. For lipid, can use a combination of Nile red staining and DAPI to count lipid producing cells and total cells in field of view when counting DAPI stained cells.**

I do believe quantitating mineral and lipid would be beneficial and can be done. Especially if it is to be published in this journal. I think this is vital for experiments concerning G9a. Knocking down or deletion of G9a drastically affects proliferation and promotes PCD. This skews the results when visually looking at mineral or fat. Therefore it has to be quantified and made relative to cell number or DNA content per well. Once cells form required mineral, researchers can add 0.6M HCl to the well for 2 hours. Then measure the dissolved mineral in the HCl using the mineral stain Calcium Arsenazo111 from Pointe Scientific. The Cells that remain behind in the well can be treated with PBS/proteinase at 56 degrees for one hour and DNA quantitated using Pico green. I routinely do this on BMSC, DPSC, adipose derived stem cells.

Lipid can be easily stained with Nile red for 15 minutes and then counterstained with DAPI. Pictures can be taken on fluorescent microscope for Nile red positive cells and DAPI stained cells. Nile red positive/DAPI positive multiplied by 100 would give percentage.

They can do the above for the most critical experimentor maybe show that when G9a or PTEN is manipulated in their hands, there is not a dramatic change in cell number?

Response:

Thank you very much for your comments that allow us to greatly improve the quality of our manuscript. We have now revised the manuscript according to your comments.

For quantification of adipogenesis, the cells were stained with Nile red and DAPI at the same time. The data were shown as Nile red positive/DAPI positive multiplied by 100 (Nile red positive cell percentage). The new data are shown in Figure 1A, 3B, 3E, 4B, 5F and 6F.

For osteogenesis, the calcium contents were quantified relative to DNA contents. In brief, the calcium contents were measured using Arsenazo III kit, while the DNA contents of the cells that remained behind in the well were quantitated using

Picogreen fluorescent dye. Calcium content was expressed as micrograms of Ca^{2+} / μg DNA. The new data are shown in Figure 1C, 3C, 3F, 4D, 5G and 6G.

The detailed methods were shown in the Materials and methods in the revised version as bellows:

Nile red staining

For Nile red staining, medium was aspirated and washed twice with PBS. Cells were then fixed in 1 ml of 10 % neutral-buffered formalin for 60 min. Following fixation, cells were washed once with PBS and then incubated for 15 min with 250 $\mu\text{g}/\text{ml}$ Nile red (Sigma) in PBS at room temperature. The cells were counterstained with 4',6-diamidino-2-phenylindole (DAPI), and images were captured using digital imaging ($\times 200$). Nile red positive cell percentages were counted.

Calcium assay

Calcium contents of cells were quantified by colorimetric endpoint assay based on the complexation of Ca^{2+} with Arsenazo III molecule at 1:2 ratio to a blue-purple product. Aliquots of 100 μl of Arsenazo III solution (Pointe Scientific, Canton, MI) were added to the wells and incubated for 10 min at room temperature. After removal of the solution, aliquots of 100 μl of 0.6M HCl were added to the wells for 2 h at room temperature. The dissolved mineral in the HCl solution was quantified spectrophotometrically at 650 nm. A standard dilution series of calcium standard (Merck, Billerica, MA) ranging from 0 to 160 $\mu\text{g}/\text{ml}$ was prepared and quantified spectrophotometrically at 650 nm. The amount of DNA in the wells was determined with Picogreen fluorescent dye (Molecular Probes, Eugene, OR) according to the manufacturer's instructions. Calcium content was expressed as micrograms of Ca^{2+} / μg DNA.

As for reviewer 1 comments in regards to age of cells, it is a valid point however most experiments are examining treated versus control within the same population of cells so it becomes somewhat a redundant point. I think it is adequately addressed.”

Response:

Thank you very much.

Reviewer #3 (Remarks to the Author):

About the manuscript NCOMMS-18-00244 submitted to Nature Communications, entitled "PTEN activation and methylation are associated with lineage commitment and imply tumorigenesis in mesenchymal stem cell" by Prof HUNG and colleagues (previous title: "PTEN methylation is associated with lineage commitment and tumorigenesis in adult stem cells"), that presents a study and characterization of human adult mesenchymal stem cells (MSCs) isolated from dental pulp (DP) and from bone marrow (BM): I reviewed the responses to my questions and the corrections introduced in the manuscript and, in my opinion, the article has greatly improved and can now be accepted for publication.

Response:

Thank you very much.

REVIEWERS' COMMENTS:

Reviewer #2 (Remarks to the Author):

My coconcerns have been addressed. Well done.